# Model Merging is Secretly Certifiable: Non-Vacuous Generalisation Bounds for Low-Shot Learning

## Abstract

Certifying the IID generalisation ability of deep networks is the first of many requirements for trusting AI in high-stakes applications from medicine to security. However, when instantiating generalisation bounds for deep networks it remains challenging to obtain non-vacuous guarantees, especially when applying contemporary large models on the small scale data prevalent in such high-stakes fields. In this paper, we draw a novel connection between a family of learning methods based on model fusion and generalisation certificates, and surprisingly show that with minor adjustment several existing learning strategies already provide non-trivial generalisation guarantees. Essentially, by focusing on data-driven learning of downstream tasks by fusion rather than fine-tuning, the certified generalisation gap becomes tiny and independent of the base network size, facilitating its certification. Our results show for the first time non-trivial generalisation guarantees for learning with as low as 100 examples, while using vision models such as VIT-B and language models such as mistral-7B. This observation is significant as it has immediate implications for facilitating the certification of existing systems as trustworthy, and opens up new directions for research at the intersection of practice and theory.

## 1 Introduction

AI is now pervasively applied across industry and society. As it increasingly impacts high-stakes applications, this raises a series of questions about to what extent we can rely on the predictions produced by AI. While there are many facets of trustworthiness—such as explainability, fairness, adversarial robustness, etc (Mucsányi et al., 2023)—the most fundamental requirement is generalisation from training data to testing data of the same distribution. *Can minimum generalisation accuracy be guaranteed, so that application domain experts can assess whether a model is performant enough to be relied upon?*

This question has traditionally been approached either empirically, by evaluation on large held out test sets, or by learning theoretic analysis that produces generalisation bounds guaranteeing test performance mathematically (Shalev-Shwartz & Ben-David, 2014). Unfortunately, such mathematical guarantees have not been very successful in the deep learning era, because instantiating such bounds for large neural networks usually leads to vacuous certificates (Neyshabur et al., 2017; Jiang et al., 2020). This is essentially because the looseness of the bound is proportional to the model's capacity, which is vast for neural networks Zhang et al. (2017). The challenge of producing practically relevant generalisation bounds is further exacerbated, because the high-stakes applications where guarantees are the most crucial (e.g., in the medical and security domains) are also often those with comparatively limited amounts of training data (Song et al., 2023). The looseness of the generalisation guarantee is usually inversely dependent on the training set size, making low-shot scenarios particularly challenging to certify (Shalev-Shwartz & Ben-David, 2014). Moreover, standard empirical validation techniques often fail in the low-data regime (Shimabucoro et al., 2024). Despite theoretical progress in deriving improved generalisation bounds (Rothfuss et al., 2022), to our knowledge there are no successful cases of instantiating non-vacuous certificates for contemporary large neural networks in low-shot scenarios.

In this paper we do not attempt to derive new theory or algorithms, but rather draw a novel connection between classic learning theoretic bounds and the increasingly popular family of algorithms built on model merging (Goddard et al., 2024), also known as model fusion (Li et al., 2023). We show that, with minor modification, a careful application of existing model merge learners can already achieve non-trivial generalisation bounds, *even in the few-shot learning regime*. Model merging can be seen as a type of multi-source Lee et al. (2019) transfer learning (Pan & Yang, 2009), where multiple pre-trained source models are assumed available, and the target problem is solved by learning a weighted fusion of available source models. The key intuition is that the number of learnable parameters is dependent on the number of source models, and potentially independent of their size. Thus learning can be parameter-light no matter the size of the source models to merge. We show that such merging learners can lead to meaningful generalisation guarantees, even for large source models that would be impossible to certify by learning directly.

More specifically, our analysis and evaluation shows several novel outcomes: (i) Some simple model merging learners can achieve non-trivial guarantees almost off-the-shelf. (ii) These learners can lead to guarantees when instantiated in multiple types of standard bound (both Gaussian distributed PAC Bayes and discretization bounds). (iii) Modifying the model merge objective to optimize the PAC Bayes generalisation bound, rather than solely the training set likelihood, enables more sophisticated model merge learners to also achieve non-trivial guarantees and leads to best overall combination of practical and guaranteed test performance. (iv) The first demonstration of non-vacuous generalisation guarantees with large models (VIT-B, and mistral-7B) in the low-shot regime.

## 2 RELATED WORK

**Model Merging**    A recently popular family of multi-source transfer learning methods is model merging Li et al. (2023), which has gained attention as many leaderboard-topping LLMs are merges of other models Goddard et al. (2024). Model merging can be used for both creating a multi-task model with many different capabilities Davari & Belilovsky (2024), as well as learning the best possible model for one specific task of interest Akiba et al. (2025). Early merging methods such as task arithmetic Ilharco et al. (2023) showed that learning simple model-wise parameter-space linear combinations of existing models can be effective, while Lora HUB Huang et al. (2023) showed that learning model-wise parameter-space linear combinations of pre-trained LORA adapters Hu et al. (2021) can be highly effective for solving downstream tasks. In these cases the learnable parameters for the downstream tasks are a handful of weighting factors compared to billions of parameters in the source models themselves. More sophisticated methods in this family developed richer representations for learning such as layer-wise Yang et al. (2024) merge weighting, improved optimisation objectives for merging such as fisher information Matena & Raffel (2022), and other improvements such as resolving sign disagreements between source model weights Yadav et al. (2024).

In this paper we focus on model merging algorithms which yield a new model by linearly interpolating the weights of multiple pre-existing networks. More specifically, given $k$ models with parameters $\theta_1, \cdots, \theta_k$, a merged model $\theta_m$ is obtained as

$$\theta_m = \alpha_1 \cdot \theta_1 + \cdots + \alpha_k \cdot \theta_k,$$

where $\alpha_1, \cdots, \alpha_k$ are merging weights. Such linear combinations can be carried out at different granularities: model-wise Ilharco et al. (2023); Huang et al. (2023) (treating $\theta_k$ as an entire network), layer-wise Yang et al. (2024) (assigning different weights per block), or weight-wise Matena & Raffel (2022) (combining individual scalar parameters). In our scope we do not consider weight-wise approaches, since we rely on learning only a small number of $\alpha_k$ parameters in total. With this parameterization, the learning stage consists of minimizing the training loss with respect to $\alpha_k$, and in the testing stage inference is performed using the single merged $\theta_m$. The main contribution of this paper is to show that several methods from this family with low-dimensional learnable representations can be straightforwardly certified to have meaningful generalisation guarantees, even in the low-data regime.

**Model Certification**    Guaranteeing the generalisation error rate for machine learning models has been a long-standing endeavour Shalev-Shwartz & Ben-David (2014) both for academic understanding and highly practical reasons. Typical learning theoretic generalisation guarantees upper bound the test error in terms of the training error plus a certified generalisation gap term that essentially

describes how much the learner could possibly have overfit to the training set. While theoretical work in bound development has been extensive, attempts to instantiate bounds to generate concrete guarantees for deep neural networks usually lead to vacuous certificates with no practical impact. This is essentially because large neural networks can fit random labels Zhang et al. (2017) and it is hard to meaningfully shrink the certified generalisation gap term. Nevertheless there have been a few successful at instantiating meaningful concrete guarantees. The seminal work of Perez-Ortiz et al. (2021) demonstrated non-vacuous certificates with classic PAC Bayes bounds at the scale of 10M parameter CNNs applied to CIFAR-10, relying on 10,000s of training examples. Recently Lotfi et al. (2024) demonstrated non-vacuous bounds for 100M parameter scale LLM learning on 10,000s of datapoints via the innovative idea of discretizing weight-space to exploit finite hypothesis class bounds. Nevertheless, these few cases have all relied on substantial training sets. To our knowledge, no meaningful certificates have been demonstrated for learning large neural networks in the low-shot regime. This paper shows that this capability has been hiding in plain sight, as several model merge learners can already be certified almost off-the-shelf. Even 7B-scale LLMs can in some cases be non-vacuously guaranteed with as few as 100 examples.

## 3 CERTIFIABLE FEW-SHOT LEARNING

Few-Shot Learning (FSL) is the problem of using a small training dataset to fit a model that will generalise to novel data points. We investigate the more challenging problem of *Certifiable Few-Shot Learning*. Under this setting, conventional model training and validation pipelines would require one to split the small *support set* of available data into two smaller sets for training and validation. However, this is impractical as even a small reduction in the amount of training data will lead to a noticeable drop in model quality, and there will also not be enough validation data available to reliably assess the performance of the model. In fact, it has been shown that standard validation approaches fail catastrophically in this setting (Shimabucoro et al., 2024). The problem of certifiable FSL is therefore to both train and validate a model using only the small support set.

For a few-shot learning problem we have a support set, $S = \{(\mathbf{x}_i, \mathbf{y}_i)\}_{i=1}^n$, that a learner can use to fit the model parameters, $\boldsymbol{\theta}$. The quality of the model parameters can be estimated using the support set via the empirical risk, denoted as

$$\hat{L}(\boldsymbol{\theta}) = \frac{1}{n} \sum_{i=1}^n \ell(f_{\boldsymbol{\theta}}(\mathbf{x}_i), \mathbf{y}_i). \tag{1}$$

In practice, if the same data is used for both fitting the model and estimating the performance of the model, we will have an overly optimistic picture of how effective the model is. The quantity that we would ideally like to measure is the population risk,

$$L(\boldsymbol{\theta}) = \mathbb{E}_{(\mathbf{x}, \mathbf{y})}[\ell(f_{\boldsymbol{\theta}}(\mathbf{x}), \mathbf{y})], \tag{2}$$

and most work on few-shot learning will estimate this quantity using an independent *query set* that is several times larger than the support set. This provides an unbiased estimate of the population risk, and when the query set is reasonably large (e.g., a few hundred) the confidence interval around this estimate can have negligible width (Foong et al., 2021).

The assumption that there exists a large query set for validation is unlikely to be true if one is in a situation where only a small set of training data is available. An alternative approach to estimating the population risk is to instead consider a quantity known as the generalisation gap,

$$G(\boldsymbol{\theta}) = L(\boldsymbol{\theta}) - \hat{L}(\boldsymbol{\theta}). \tag{3}$$

The population risk can then be estimated by adding this to the empirical risk,

$$L(\boldsymbol{\theta}) = \hat{L}(\boldsymbol{\theta}) + G(\boldsymbol{\theta}). \tag{4}$$

More commonly, we will have an upper bound on the generalisation gap, $\bar{G}(\theta) \geq G(\theta)$, that holds in expectation over the support set. In this case, we can obtain a probabilistic upper bound on the population risk. Much like a confidence interval one might obtain from a query set, the probability that this holds can be directly controlled by the user. The resulting bound is a conservative estimate of the population risk,

$$L(\boldsymbol{\theta}) \leq \hat{L}(\boldsymbol{\theta}) + \bar{G}(\boldsymbol{\theta}). \tag{5}$$

Techniques for developing upper bounds on the generalisation gap include classic methods, such as the VC dimension, more modern methods like Rademacher complexity, and the state-of-the-art PAC-Bayes framework that we focus on in this paper. See Shalev-Shwartz & Ben-David (2014) for more details about all of these approaches.

### 3.1 PAC-BAYES GENERALISATION BOUNDS

Historically, it has been the case that upper bounds on the generalisation gap for deep neural networks have been very loose, leading to vacuous certifications; i.e., they are so pessimistic that they give meaningless guarantees, such as the error being less than 200%. However, recent work has shown that it is possible to obtain non-vacuous certifications on image classification problems with relatively small networks and sufficiently large training datasets (Dziugaite & Roy, 2017; Perez-Ortiz et al., 2021), and even transformer-based language models (Lotfi et al., 2024). A common ingredient in all of these recent works on obtaining non-vacuous certifications is the use of the PAC-Bayes framework.

There are several subtleties involved in leveraging PAC-Bayes bounds for certification of conventional deep networks. First, they do not provide a certification for a single $\boldsymbol{\theta}$ produced by a learner. Instead, they require that the learner produce a distribution, $Q$, over parameters. Each time a prediction is to be made, one first samples a $\boldsymbol{\theta}$ from $Q$, and then a prediction is made with that single choice of $\boldsymbol{\theta}$. Crucially, a new set of model parameters is sampled for each new data point for which a prediction will be made. $Q$ is often referred to as the PAC-Bayes posterior, which should not be confused with the Bayesian posterior. The tightness of the certification depends substantially on the Kullback-Leibler divergence between $Q$ and the so-called PAC-Bayes prior, $P$. Similarly to the Bayesian case, the prior cannot depend on the data used to fit the PAC-Bayes posterior but can otherwise be chosen to be any distribution over the model parameters.

When the learner produces a distribution over parameters rather than a single $\boldsymbol{\theta}$, we denote the empirical risk and population risk as

$$\hat{L}(Q) = \mathbb{E}_{\boldsymbol{\theta} \sim Q}[\hat{L}(\boldsymbol{\theta})] \qquad \text{and} \qquad L(Q) = \mathbb{E}_{\boldsymbol{\theta} \sim Q}[L(\boldsymbol{\theta})], \tag{6}$$

respectively.

**The Conventional Bound**  A large number of PAC-Bayes generalisation bounds have been developed (we refer the reader to Alquier (2021) for an overview), but we consider two in particular. The first of these is slightly looser but easy to compute.

**Theorem 1.** *For any PAC-Bayes posterior, Q, and PAC-Bayes prior, P, we have with confidence at least $1 - \delta$ that*

$$L(Q) \leq \hat{L}(Q) + \sqrt{\frac{\mathrm{KL}(Q\|P) + \ln(n/\delta)}{2(n-1)}}. \tag{7}$$

**The Seeger Bound**  The second PAC-Bayes bound we consider is the bound in Langford & Seeger (2001) and Seeger (2002) which can provide better guarantees when the learner is able to achieve small error on the support while still keeping the PAC-Bayes prior and posterior relatively close.

**Theorem 2** (Langford & Seeger (2001); Seeger (2002)). *For any PAC-Bayes posterior, Q, and PAC-Bayes prior, P, we have with confidence at least $1 - \delta$ that*

$$\mathrm{kl}(\hat{L}(Q)\|L(Q)) \leq \frac{\mathrm{KL}(Q\|P) + \ln(n/\delta)}{n-1},$$

*where $\mathrm{kl}(\cdot\|\cdot)$ is the equation for the KL divergence between two Bernoulli random variables in terms of their p parameters,*

$$\mathrm{kl}(p\|q) = p \ln \frac{p}{q} + (1-p) \ln \frac{1-p}{1-q}. \tag{8}$$

The main challenge involved with using this bound is appropriately choosing the family of distributions that $Q$ and $P$ come from. One needs to ensure that $Q$ can be chosen by the learner such that a significant amount of the probability mass can be assigned to well-performing models. Simultaneously, one must choose $P$ independently of the training set, but in such a way that it assigns enough mass to good models that the KL divergence is small. Finally, these choices must also allow one to compute the KL divergence so a certificate can actually be obtained. Popular families of modeling

prior and posterior for tractable computation is described in Appendix B. In the second bound, there is also the added complication that the generalisation bound does not immediately give rise to closed form expression for obtaining a performance certification. One must instead use a placeholder for $L(Q)$ and consider the equality condition of the inequality,

$$\mathrm{kl}(\hat{L}(Q)\|C) = \frac{\mathrm{KL}(Q\|P) + \ln(n/\delta)}{n - 1}. \tag{9}$$

Numerically solving this equation through Newton's method provides a value, $C$, that satisfies $L(Q) \leq C$, thus giving us a performance certificate. There are various methods for constructing upper bounds to $C$ that end up producing other (looser) PAC-Bayes bounds as special cases of this bound. We refer the reader to Dziugaite & Roy (2017) for details.

**The Test Set Bound**    A principled data-driven approach for model certification is the test set bound of Langford (2005). This is usually derived as a Chernoff upper bound on the expected value of a binomial random variables, that holds with a user-defined level of confidence.

## 4    PAC-Bayesian Model Merging

### 4.1    Interpreting off-the-shelf model mergers in terms of PAC-Bayes

We first consider *model merging* from a simple, nearly off-the-shelf perspective, where the merging procedure is directly interpreted in a PAC-Bayesian framework. Specifically, we posit a Gaussian prior and posterior with fixed variance over merging parameters:

$$\text{Prior: } \mathcal{N}(\boldsymbol{\mu}_P, \lambda_1 I), \quad \text{Posterior: } \mathcal{N}(\boldsymbol{\mu}_Q, \lambda_2 I).$$

The prior mean $\boldsymbol{\mu}_P$ is set to be uniform across the $M$ models being merged—i.e., each element of $\boldsymbol{\mu}_P$ is $\frac{1}{M}$. Once the posterior mean $\boldsymbol{\mu}_Q$ is found by *any* merging procedure (for instance, a conventional gradient-based or gradient-free learner), we can *re-interpret* the result as the posterior mean in the above PAC-Bayesian setup. Note that utilising Gaussian distribution is just modeling choice, not an assumption about parameter distribution. The PAC-Bayes theorem holds for 'any' prior and posterior distributions, and they do not need to be Gaussian. However, to instantiate the bound, we need to make a modeling choice amenable to computing KLDs. We chose Gaussian for convenience, following common practice in PAC Bayes, any other suitable distribution could also be used.

**Secretly Non-Vacuous Bounds.**    Under this viewpoint, existing merging method can be "secretly" providing a non-vacuous PAC-Bayesian bound without any modification. Indeed, once we plug the merging parameters back into the posterior $\mathcal{N}(\boldsymbol{\mu}_Q, \lambda_2 I)$, we can evaluate the corresponding PAC-Bayes bound. This requires no change to the existing merging algorithm—we simply measure the empirical loss of the "posterior" (the merged model) and compute the KL term $\mathrm{KL}(Q \| P)$ based on the prior. Hence, *off-the-shelf* model merging algorithms can be equipped with PAC-Bayes analyses at no extra cost, potentially revealing that they already achieve non-vacuous bounds in practice since their number of parameters tends to be tiny, regardless of network size.

### 4.2    PAC-Bayes Informed Learning Objectives (Direct Optimisation)

Although the re-interpretation in Section 4.1 can yield meaningful bounds with zero modifications to the merging procedure, in certain scenarios these bounds may be *vacuous* (e.g., when there is a certain large number of merging parameters). To address this, we propose *directly optimising* a PAC-Bayesian objective to make the bound tighter.

Specifically, we minimise the upper bound of $L(Q)$ in Theorem 2 as same as Dziugaite & Roy (2017):

$$\min_{\boldsymbol{\mu}_Q \in \mathbb{R}^d} \hat{L}(Q) + \sqrt{\frac{\mathrm{KL}(Q\|P) + \log \frac{n}{\delta}}{2(n - 1)}}, \tag{10}$$

where $\hat{L}(Q)$ is the empirical 0-1 loss of the posterior distribution $Q$. In practice, the 0-1 loss is non-differentiable, so we resort to gradient-free methods (e.g., evolutionary search). However, *any* strategy that optimises this objective in parameter space can make the posterior "fit" the PAC-Bayes bound more tightly.

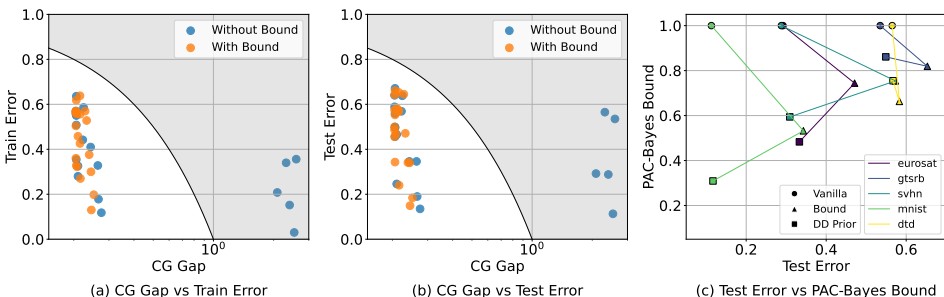

(a) CG Gap vs Train Error     (b) CG Gap vs Test Error     (c) Test Error vs PAC-Bayes Bound

Figure 1: Impact of learning with bound-optimization and data-dependent prior. CLIP-ViT-B/32 on EuroSAT, GTSRB, SVHN, MNIST and DTD. Each point corresponds to a different combination of dataset and merging algorithm. CG Gap refers to the certified generalisation gap. (a) Certified Generalisation Gap vs. Train Error, (b) Certified Generalisation Gap vs. Test Error. Bound optimization is often crucial for non-vacuous results (white zone). (c) Test Error vs. PAC-Bayes Bound. Combining bound optimization with DD prior leads to a good tradeoff for LW-Adamerge.

**Easily Upgrading Merging Algorithms.** Notably, this approach requires only minor extensions to Section 4.1. Instead of using an existing merging procedure to optimize solely $\hat{L}$, one can incorporate the PAC-Bayes term (the second term in Eq. 10) into the objective and *upgrade* the merging algorithm to explicitly optimise for tighter PAC-Bayesian guarantees. Our experiments show that where the off-the-shelf mergers in Section 4.1 yield a vacuous bound, this upgrade can often improve vacuous certificates to become *non-vacuous* guarantees, with little extra effort. This demonstrates the versatility and robustness of the proposed PAC-Bayesian model merging framework.

**Data-Dependent Priors.** One way to obtain a tighter bound is to refine the prior by leveraging part of the training data (Perez-Ortiz et al., 2021). An *uninformative* prior risks yielding a large KL divergence if there is no posterior close to it that well explains the data, thus making the overall bound less tight. To mitigate this, we can split the training set and use a portion of it to perform a standard merging procedure first, treating the resulting merged parameters as a data-dependent prior mean. We then apply the PAC-Bayes optimisation (Eq. 10) on the remaining data. This two-stage process can substantially improve the tightness of the final bound, as the prior is now more likely to be near a good posterior.

## 5 EXPERIMENTS

### 5.1 EXPERIMENTAL SETUPS

We conduct experiments on two tasks—image classification and text classification—using four merging methods: Task Arithmetic Ilharco et al. (2023), Ties-Merging Yadav et al. (2024), Task-wise Adamerging Yang et al. (2024), and Layer-wise Adamerging Yang et al. (2024). For image classification, we employ CLIP-ViT-B/32 Dosovitskiy (2020); Radford et al. (2021) (approximately 88M parameters) and its variant trained on eight datasets: Stanford Cars Krause et al. (2013), EuroSAT Helber et al. (2019), RESISC45 Cheng et al. (2017), GTSRB Stallkamp et al. (2012), SUN397 Xiao et al. (2010), SVHN Netzer et al. (2011), MNIST LeCun et al. (1998), and DTD Cimpoi et al. (2014). With this pool of models, we perform hold-one-out learning, attempting to solve each held out task by merging the other models using a low-shot training set of the held out model. The main paper reports results on the five datasets that lead to succesful merge-based learning in this hold-out setup – namely, EuroSAT Helber et al. (2019), GTSRB Stallkamp et al. (2012), SVHN Netzer et al. (2011), MNIST LeCun et al. (1998), and DTD Cimpoi et al. (2014). For text classification, we utilize Mistral-7B Jiang et al. (2023) and its variants: MetaMath-Mistral-7B Met (b), Dolphin-2.1-Mistral-7B Dol and Speechless-Code-Mistral-7B-v1.0 Spe on BBH Suzgun et al. (2022) benchmark. Similarly to the vision experiments, we report certificates on the subset of datasets within BBH benchmark, where merge learning succeeds (improves on the base LLM). Note that the excluded cases correspond to failures of the base learning algorithm, not our certification strategy. Our contribution is to certify successful cases of a 3rd party learning algorithm, and there is simply no success to certify in those cases. The full results for all tasks are nevertheless given in Appendix C.

In all experiments, we train with 100 labeled examples from the target dataset, and parameterise both the PAC-Bayes priors and posteriors by diagonal Gaussian distributions with variance 0.05 for

Table 1: Test Error, Train Error, PAC-Bayes Bound, PAC-Bayes Upper Bound across 5 image classification datasets and 4 merging algorithms fusing seven Clip-ViT-B/32 derivatives. We use 100 examples from each target dataset to fit the merging parameters, combining a pool of models except the model that fine-tuned on the target dataset. Shaded cells indicate regions with vacuous bounds.

| | Metric | Original Objective | | | | | Bound Optimization | | | | |
|---|---|---|---|---|---|---|---|---|---|---|---|
| | | eurosat | gtsrb | svhn | mnist | dtd | eurosat | gtsrb | svhn | mnist | dtd |
| Task Arith. | Test Error | 0.456 | 0.671 | 0.589 | 0.246 | 0.499 | 0.456 | 0.661 | 0.575 | 0.240 | 0.500 |
| | Train Error | 0.508 | 0.636 | 0.558 | 0.280 | 0.354 | 0.504 | 0.618 | 0.568 | 0.270 | 0.360 |
| | PB Bound | 0.704 | 0.812 | 0.747 | 0.486 | 0.559 | 0.700 | 0.797 | 0.755 | 0.481 | 0.564 |
| | Upper Bound | 0.714 | 0.842 | 0.763 | 0.490 | 0.560 | 0.709 | 0.824 | 0.773 | 0.486 | 0.565 |
| Ties Merge | Test Error | 0.496 | 0.640 | 0.559 | 0.190 | 0.466 | 0.486 | 0.642 | 0.553 | 0.184 | 0.467 |
| | Train Error | 0.568 | 0.570 | 0.550 | 0.178 | 0.326 | 0.560 | 0.570 | 0.558 | 0.198 | 0.322 |
| | PB Bound | 0.756 | **0.757** | 0.741 | 0.428 | 0.534 | 0.749 | **0.757** | 0.748 | 0.437 | **0.528** |
| | Upper Bound | 0.774 | 0.775 | 0.756 | 0.444 | 0.536 | 0.765 | 0.775 | 0.764 | 0.450 | 0.530 |
| TW-Ada | Test Error | 0.346 | 0.638 | 0.346 | 0.135 | 0.569 | 0.341 | 0.645 | 0.340 | 0.149 | 0.568 |
| | Train Error | 0.410 | 0.586 | 0.328 | 0.118 | 0.442 | 0.376 | 0.570 | 0.300 | 0.130 | 0.426 |
| | PB Bound | 0.649 | 0.785 | 0.590 | 0.360 | 0.659 | **0.613** | 0.776 | **0.541** | **0.345** | 0.638 |
| | Upper Bound | 0.653 | 0.810 | 0.592 | 0.393 | 0.664 | 0.615 | 0.798 | 0.544 | 0.375 | 0.641 |
| LW-Ada | Test Error | 0.292 | 0.535 | 0.288 | 0.113 | 0.565 | 0.471 | 0.653 | 0.573 | 0.343 | 0.583 |
| | Train Error | 0.208 | 0.356 | 0.152 | 0.030 | 0.340 | 0.528 | 0.638 | 0.556 | 0.328 | 0.458 |
| | PB Bound | 1.000 | 1.000 | 1.000 | 1.000 | 1.000 | 0.744 | 0.819 | 0.754 | 0.532 | 0.663 |
| | Upper Bound | 2.295 | 2.952 | 2.558 | 2.567 | 2.651 | 0.760 | 0.853 | 0.771 | 0.534 | 0.668 |

each parameter. Any other experimental details including number of parameters for each method and hyperparameters are described in Appendix A.

## 5.2 MODEL MERGING LEARNERS CAN PROVIDE NON-VACUOUS CERTIFICATES

**Model Merging for Certification** We investigate the tightness of each generalisation bound by comparing them to an oracle test error computed using large query sets available for the image recognition tasks. Table 1 and Figure 1 summarise the results. The first observation is that in the left half of the Table 1, for three out of the four model merging methods, the generalisation bounds provide non-vacuous performance guarantees off-the-shelf. Recall that in these cases the methods have only undergone a small adaptation such that they produce a distribution over models, making them compatible with PAC-Bayes bounds—the model parameterisation and learning criterion remain the same. Comparing left and right halves of Table 1, shows that the guarantees tend to improve when using PAC-Bayes bound as the training signal. This is visualised in Figure 1(a) and (b), where we can see that all points trained with the bound lie inside the non-vacuous region, while several conventionally trained models provide vacuous guarantees. Moreover, with the exception of Layer-wise AdaMerging, the right half of Table 1 tells us that this does not lead to a noticeable drop in the test error of the models. This is a somewhat surprising result, as the PAC-Bayes bounds typically provide a conservative estimate of performance, and using them as an optimisation objective can lead to overly harsh regularisation, and underfitting. It is important to note that the methods in Table 1 and 2 are all variants of our proposed framework (e.g., different base merging algorithms combined with bound optimisation or different priors), and thus the table should be understood as an ablation study rather than a comparison to prior methods. In fact, in the low-shot + large neural network regime we study, all prior certification approaches (e.g., Full-Finetuning+Rademacher complexity (Gouk et al.) or SubLoRA+discretisation-based bounds (Lotfi et al., 2024)) lead to vacuous guarantees while only our methods provide non-vacuous and practically useful guarantees.

**Supporting Mergers with More Parameters** In Table 1, the results for Layer-wise AdaMerge lagged behind the other model merging approaches, due to the larger number of parameters it learns. To overcome this, we deploy Data-Dependent-Prior (DDP) approach. This can be done without violating the independence of the prior and support set by splitting the training data and using the first half to construct a data dependent prior, then performing certified FSL using the second half of the data. The results of using this technique to certify the performance of the Layer-wise AdaMerging method are given in Table 2 and Figure 1(c). The main takeaway from these results is that using a data dependent prior can lead to a substantially improved bound in some cases, as shown in Table 2. For some datasets (eg eurosat) this leads to substantially better bounds than the best in Table 1. Figure 1(c) shows the effect of Bound optimisation and DDP. The bottom left is the ideal point of both good empirical and certified error, and each color represents results on a different dataset. Beginning

Table 2: Effect of adopting a data-dependent prior (DDP) with layer-wise Adamerging Yang et al. (2024) across 5 image classification datasets with Clip-ViT-B/32. Shaded cells indicate the regions with vacuous bounds.

|  | Metric | eurosat | gtsrb | svhn | mnist | dtd |
|---|---|---|---|---|---|---|
| LW-Ada | Test Error | 0.292 | 0.535 | 0.288 | 0.113 | 0.565 |
|  | Train Error | 0.208 | 0.356 | 0.152 | 0.030 | 0.340 |
|  | PB Bound | 1.000 | 1.000 | 1.000 | 1.000 | 1.000 |
| LW-Ada +Bound | Test Error | 0.471 | 0.653 | 0.573 | 0.343 | 0.583 |
|  | Train Error | 0.528 | 0.638 | 0.556 | 0.328 | 0.458 |
|  | PB Bound | 0.744 | **0.819** | 0.754 | 0.532 | **0.663** |
| LW-Ada +DDP | Test Error | 0.333 | 0.549 | 0.309 | 0.117 | 0.567 |
|  | Train Error | 0.212 | 0.632 | 0.308 | 0.080 | 0.492 |
|  | PB Bound | **0.483** | 0.861 | **0.594** | **0.309** | 0.755 |

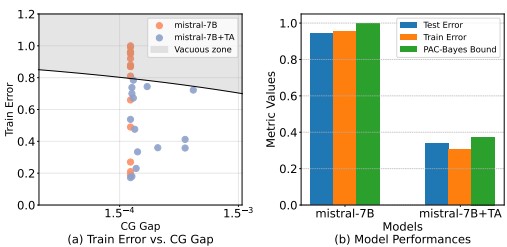

Figure 2: Comparison of mistral-7B vs. mistral-7B+Task Arithmetic (Ilharco et al., 2023). CG Gap refers to the certified generalisation gap. (a) BBH: train error vs. CG gap. Grey zone indicates vacuous region. (b) TweetEval: performance across metrics.

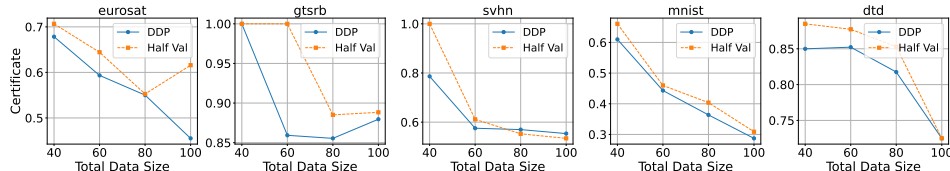

Figure 3: Change of Performance Certificates with different total data size on 5 image classification datasets with CLIP-ViT-32/B. The DDP method refers to Task-wise AdaMerging with Data-Dependent Prior. In DDP, we use half of data for fitting a prior, and the rest for fitting a posterior and bound computation. In Half Val, we used half of data for model training and the rest for computing the "test set" (confidence interval) based bound.

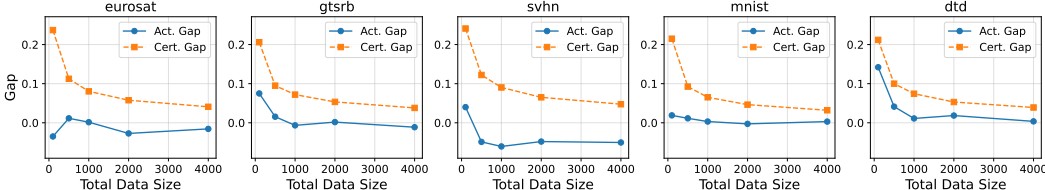

Figure 4: Change of Actual/Certified Generalisation Gap by different total data size (100, 500, 1000, 2000, 4000) on 5 image classification datasets with CLIP-ViT-32/B. Note that Act. Gap refers to Actual Generalisation Gap, computed by subtracting train error from test error, and Cert. Gap refers to Certified Generalisation Gap, computed by subtracting train error from PAC Bayes Bound.

near the top of the plot, the vanilla merging algorithms correspond to circles and exhibit vacuous generalisation guarantees. When we apply bound optimisation (triangle points), the PAC-Bayes bound decreases but test/train error might increase compared to vanilla merging since bound optimisation forces the posterior to be near the prior, which leads to underfitting. This can be resolved by using data dependent priors (square points) since it provides informative priors that enable the learning methods to find good posteriors that are still near the prior distributions. This yields improvements both on PAC-Bayes bound and Test/Train Error. In summary, it shows that both bound optimization in combination with DDP are required to simultaneously achieve good certificate and test error.

**Comparison with Test Set Bound** We also provide results comparing the proposed approach with the so-called test set bound (Sec. 3.1, Langford & Seeger (2001)), that we note actually uses a validation set. Figure 3 compares our Data-Dependent Prior (DDP) approach with the *Half Val* strategy, which uses half of the data for training and the rest for validation, for the case of Task-wise AdaMerging of CLIP-ViT-32/B models. The results show that our approach usually leads to a better performance certificate. Intuitively, this is because our data-dependent prior approach can use the full data, albeit split between prior and posterior training, resulting in reduced variance for both parameter estimators (i.e., the learning algorithm) and performance estimators (i.e., the performance certificate). In contrast, the half-train/half-val approach is only able to use half of the data for training, leading to higher variance estimators in both model training and validation.

Table 3: Certifying low-shot learning of `mistral-7B` LLM on BBH benchmark. Error & certified error for zero-shot (left) vs 100-shot TA learning (Ilharco et al. (2023), right). Shaded regions indicate vacuous bounds. TA learning leads to substantially better certificates than the Zero-Shot Test Set bound.

| Task | Vanilla Mistral-7B | | | | Task-Arithmetic | | | |
|---|---|---|---|---|---|---|---|---|
| | Test Err. | Train Err. | PB Bound | Upper Bound | Test Err. | Train Err. | PB Bound | Upper Bound |
| Logical Deduction (5 objs.) | 0.933 | 0.960 | 1.000 | 1.165 | 0.703 | 0.738 | **0.887** $(-0.113)$ | 0.944 |
| Hyperbaton | 0.260 | 0.210 | 0.402 | 0.415 | 0.239 | 0.172 | **0.357** $(-0.045)$ | 0.377 |
| Snarks | 0.641 | 0.660 | 0.829 | 0.865 | 0.392 | 0.358 | **0.603** $(-0.226)$ | 0.605 |
| Boolean Expressions | 0.247 | 0.190 | 0.379 | 0.395 | 0.247 | 0.180 | **0.368** $(-0.011)$ | 0.386 |
| Geometric Shapes | 0.920 | 0.880 | 1.000 | 1.085 | 0.757 | 0.744 | **0.897** $(-0.103)$ | 0.961 |
| Causal Judgement | 0.943 | 0.920 | 1.000 | 1.125 | 0.451 | 0.412 | **0.655** $(-0.355)$ | 0.659 |
| Reasoning Colored Objs. | 0.907 | 0.870 | 1.000 | 1.075 | 0.541 | 0.538 | **0.730** $(-0.270)$ | 0.743 |
| Formal Fallacies | 1.000 | 1.000 | 1.000 | 1.205 | 0.433 | 0.476 | **0.678** $(-0.322)$ | 0.684 |
| Tracking Shuffled Objs. | 0.927 | 0.870 | 1.000 | 1.075 | 0.747 | 0.672 | **0.840** $(-0.160)$ | 0.879 |
| Movie Recommendation | 0.560 | 0.490 | 0.687 | 0.695 | 0.424 | 0.334 | **0.543** $(-0.144)$ | 0.544 |
| Ruin Names | 0.940 | 0.990 | 1.000 | 1.195 | 0.749 | 0.722 | **0.902** $(-0.098)$ | 0.976 |
| Navigate | 0.760 | 0.810 | 1.000 | 1.015 | 0.492 | 0.360 | **0.584** $(-0.416)$ | 0.585 |
| Logical Deduction (3 objs.) | 0.973 | 0.960 | 1.000 | 1.165 | 0.705 | 0.700 | **0.860** $(-0.140)$ | 0.906 |
| Logical Deduction (7 objs.) | 0.960 | 0.950 | 1.000 | 1.155 | 0.840 | 0.784 | **0.918** $(-0.082)$ | 0.991 |
| Sports Understanding | 0.233 | 0.270 | 0.470 | 0.475 | 0.200 | 0.230 | **0.429** $(-0.041)$ | 0.439 |

**Certified Generalisation Gap with More Data**   We provide the results how data size can effect the certified generalisation gap. Figure 4 shows the Actual/Certified Generalisation Gap with varying train dataset size, for the case of Bound optimisation with Task-wise AdaMerging of CLIP-ViT-32/B models. The results show that increasing the dataset size shrinks the certified generalisation gap (i.e., the gap between PAC-Bayes Bound and Train Error becomes small). The test performance can be certified to be within $5\%$ of the training performance for all datasets. This shows that the proposed approach can yield practical, not merely weakly non-vacuous, certification of large neural networks with only a few thousand data points – still substantially less data than prior state of the art Lotfi et al. (2024) in learning theoretic work.

**Can We Scale to LLMs?**   As a proof of concept, we investigate whether one can obtain non-vacuous generalisation bounds for Large Language Models. This is explored in the context of Mistral-7B (Jiang et al., 2023), which we adapt to new tasks using the Task-Arithmetic Ilharco et al. (2023). For comparison, we include results of using the Mistral-7B base model in a zero-shot setting. We present results using the 100 examples per task on BBH benchmark Suzgun et al. (2022) in Table 3 and Figure 2(a), and using 1,000 examples on a TweetEval-Hate Speech Detection benchmark Basile et al. (2019) in Figure 2(b). In many situations the zero-shot performance of the base model is quite poor, indicating that the adaptation procedure is necessary. Overall, the results show that (i) Task-Arithmetric learning is able to make substantial gains over zero-shot application of the base model, in terms of measured test error; and importantly (ii) the PAC-Bayes generalisation bound gives non-vacuous guarantees on few-shot adaptation of the Mistral-7B Large Language Model. More results including experiments with Llama3-8B-Instruct (AI@Meta, 2024) can be found in C.

## 6   CONCLUSION

This paper explores the intersection between an emerging family of model-merging learners, and various PAC-Bayes generalisation bounds. The results show that, with minor extension to direct bound optimization objectives, several existing model-merging algorithms can be certified with non-vacuous guarantees - even when applied to learn 7B scale LLMs in the low-shot regime. This is a noteworthy advance in the both the size of successfully certified models, as well as the sparsity of the dataset for certified learning. This analysis may open up new avenues for both theoretical and algorithmic research, by identifying the promise of focusing guarantees on learners re-parametrized in terms of sparse merging weights rather than the much larger number of base neural network parameters. Moreover, it could provide immediate impact in facilitating the certification of existing AI systems where there are legal or ethical requirements to do so.

**Reproducibility statement**   We aim to make our experiments fully reproducible. Details of the methodology and training setup are provided in Sections 4, 5.1, and A. All datasets used in this work are publicly available. The code, including training and evaluation scripts as well as configuration files specifying hyperparameters and random seeds, will be released upon publication.

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

## A    ADDITIONAL EXPERIMENTAL SETUP

### A.1    EXPERIMENTAL DETAILS

For Task Arithmetic (Ilharco et al., 2023) and Ties-Merging (Yadav et al., 2024), they use only one parameter that will be multiplied to the average of task vectors and then the multiplied vector will be added to the base model. For Task-wise AdaMerging (Yang et al., 2024), the number of model parameters is same as the number of models being merged, which is 7 for CLIP-ViT-B32 experiments. For layer-wise AdaMerging (Yang et al., 2024), the number of parameters becomes multiplication of number of models and number of layers, which is $7 \times 28 = 196$ for CLIP-ViT-B32. $\delta$ is set to 0.05. All the experiments were performed on a single NVIDIA-V100 GPU using the Fusion-Bench Tang et al. (2024) code base. For PAC-Bayes bound optimization, we use CMA-ES algorithm implemented by Rapin & Teytaud (2018) as per Huang et al. (2023) with all hyperparameters set to their default values, including initial scale as 1 and population size $4 + 3\ln(n)$. Any details that are different from above are specified in each experiment.

### A.2    EVALUATION METRICS

**Train and Test Error**    Because we use randomised classifiers, train error, $\hat{L}(Q)$, and test error, an estimate of $L(Q)$, were computed using Monte-Carlo approximation using 10 sampled weights from posterior. Test error was computed on whole test set and train error was computed on train set used for training with the merging algorithms.

**PAC-Bayes (PB) and Upper Bound**    PAC-Bayes Bound refers to the value numerically computed following Theorem 2, which acts as the upper bound of test error $L(Q)$. Upper bounds means the upper bound of PAC-Bayes bound computed by Eq. 10. Throughout the paper, "vacuous" refers to the situation when PAC-Bayes bound is larger than 1, thus guaranteeing nothing.

**Certified Generalisation Gap**    PAC-Bayesian theory bounds the test error $L(Q)$ by the sum of the train error and the certified generalisation gap term. The certified generalisation gap term simply corresponds to the third term in Theorem 1, if Theorem 1 is used. In practice, we use the slightly more complex Theorem 2 because it provides tighter bound, but also requires numerically inverting the KL divergence between Bernoulli distributions. Hence our certified generalisation gap term is computed by subtracting train error $\hat{L}(Q)$ from certified performance computed by this numerical inversion process. Since the other terms influence certified generalisation gap term are determined by fixed values $(m, \delta)$, certified generalisation gap term becomes equivalent to KL-Divergence term.

## B    DISTRIBUTION FAMILIES FOR TRACTABLE COMPUTATION

Multivariate Gaussians are a popular choice of distribution families used in existing work on certifying models in the case where large training datasets are available. If the prior is chosen with a large enough variance then the KL divergence with a relatively concentrated posterior does become too large. Moreover, it is possible to write down a closed-form expression for compute the KL divergence between two multivariate Gaussians. When the Gaussians are also restricted to having diagonal—or even scaled identity—covariance matrices the number of additional parameters that must be fit is only increased by a factor of at most two, and the KL divergence is very efficient to compute. In particular, for two Gaussians with means, $\boldsymbol{\mu}_1$ and $\boldsymbol{\mu}_2$, and covariances, $\sigma_1^2 I$ and $\sigma_2^2 I$, the KL divergence is given by

$$\mathrm{KL}(\mathcal{N}(\boldsymbol{\mu}_1, \sigma_1^2 I) \| \mathcal{N}(\boldsymbol{\mu}_2, \sigma_2^2 I)) = \frac{d}{2}\left(\frac{\sigma_1^2}{\sigma_2^2} - 1 - \ln\frac{\sigma_1^2}{\sigma_2^2}\right) + \frac{1}{2\sigma_2^2}\|\boldsymbol{\mu}_1 - \boldsymbol{\mu}_2\|^2 \qquad (11)$$

Another family that is less widely used in the literature is that of categorical distributions. These distributions must necessarily be over a finite set of potential model parameter settings specified before the support set is seen. In many situations this is a prohibitive requirement, but we will see that in the case of model merging this is not fatal. A special case worth considering is when the prior is chosen to be the uniform distribution. In this case, the KL divergence between the uniform

Table 4: Performance metrics (Test Error, Train Error, PAC-Bayes Bound, Upper Bound) across 8 image classification datasets with Clip-ViT-B/32. We use 100 examples from each target dataset to fit the merging parameters, combining a pool of models fine-tuned on all other datasets. Rows with "+ Bound" indicate that we use the PAC-Bayes bound as the optimisation objective. We also provide an additional "Upper Bound" row for each method, which indicates the upper bound of PAC-Bayes bound before numerical computation of exact PAC-Bayes bound. The bold faced number indicates the best PAC-Bayes bound on each dataset. The shaded columns are the datasets where the underlying merging schemes failed to attain meaningful performance gains.

| Method | Metric | cars | eurosat | resisc45 | gtsrb | sun397 | svhn | mnist | dtd |
|---|---|---|---|---|---|---|---|---|---|
| Zero-Shot | Test Error | 0.401 | 0.538 | 0.393 | 0.675 | 0.369 | 0.684 | 0.518 | 0.567 |
| | Train Error | 0.450 | 0.600 | 0.450 | 0.660 | 0.330 | 0.610 | 0.490 | 0.450 |
| | PAC-Bayes Bound | 0.651 | 0.782 | 0.651 | 0.830 | **0.533** | 0.790 | 0.687 | 0.651 |
| | Upper Bound | 0.655 | 0.805 | 0.655 | 0.865 | 0.535 | 0.815 | 0.695 | 0.655 |
| Task Arithmetic | Test Error | 0.409 | 0.456 | 0.393 | 0.671 | 0.374 | 0.589 | 0.246 | 0.499 |
| | Train Error | 0.444 | 0.508 | 0.458 | 0.636 | 0.352 | 0.558 | 0.280 | 0.354 |
| | PAC-Bayes Bound | 0.646 | 0.704 | 0.659 | 0.812 | 0.556 | 0.747 | 0.486 | 0.559 |
| | Upper Bound | 0.650 | 0.714 | 0.664 | 0.842 | 0.557 | 0.763 | 0.490 | 0.560 |
| Task Arithmetic + Bound | Test Error | 0.434 | 0.456 | 0.384 | 0.661 | 0.373 | 0.575 | 0.240 | 0.500 |
| | Train Error | 0.448 | 0.504 | 0.454 | 0.618 | 0.358 | 0.568 | 0.270 | 0.360 |
| | PAC-Bayes Bound | 0.649 | 0.700 | 0.655 | 0.797 | 0.563 | 0.755 | 0.481 | 0.564 |
| | Upper Bound | 0.653 | 0.709 | 0.660 | 0.824 | 0.564 | 0.773 | 0.486 | 0.565 |
| Ties-Merging | Test Error | 0.404 | 0.496 | 0.384 | 0.640 | 0.367 | 0.559 | 0.190 | 0.466 |
| | Train Error | 0.440 | 0.568 | 0.450 | 0.570 | 0.338 | 0.550 | 0.178 | 0.326 |
| | PAC-Bayes Bound | 0.642 | 0.756 | 0.651 | **0.757** | 0.542 | 0.741 | 0.428 | 0.534 |
| | Upper Bound | 0.645 | 0.774 | 0.655 | 0.775 | 0.544 | 0.756 | 0.444 | 0.536 |
| Ties-Merging + Bound | Test Error | 0.402 | 0.486 | 0.381 | 0.642 | 0.367 | 0.553 | 0.184 | 0.467 |
| | Train Error | 0.438 | 0.560 | 0.454 | 0.570 | 0.338 | 0.558 | 0.198 | 0.322 |
| | PAC-Bayes Bound | **0.640** | 0.749 | 0.655 | **0.757** | 0.542 | 0.748 | 0.437 | **0.528** |
| | Upper Bound | 0.643 | 0.765 | 0.660 | 0.775 | 0.544 | 0.764 | 0.450 | 0.530 |
| Task-wise Ada | Test Error | 0.418 | 0.346 | 0.373 | 0.638 | 0.371 | 0.346 | 0.135 | 0.569 |
| | Train Error | 0.464 | 0.410 | 0.472 | 0.586 | 0.336 | 0.328 | 0.118 | 0.442 |
| | PAC-Bayes Bound | 0.677 | 0.649 | 0.682 | 0.785 | 0.554 | 0.590 | 0.360 | 0.659 |
| | Upper Bound | 0.683 | 0.653 | 0.688 | 0.810 | 0.555 | 0.592 | 0.393 | 0.664 |
| Task-wise Ada + Bound | Test Error | 0.436 | 0.341 | 0.387 | 0.645 | 0.379 | 0.340 | 0.149 | 0.568 |
| | Train Error | 0.460 | 0.376 | 0.420 | 0.570 | 0.302 | 0.300 | 0.130 | 0.426 |
| | PAC-Bayes Bound | 0.668 | **0.613** | 0.639 | 0.776 | 0.539 | **0.541** | **0.345** | 0.638 |
| | Upper Bound | 0.673 | 0.615 | 0.642 | 0.798 | 0.541 | 0.544 | 0.375 | 0.641 |
| Layer-wise Ada | Test Error | 0.430 | 0.292 | 0.367 | 0.535 | 0.370 | 0.288 | 0.113 | 0.565 |
| | Train Error | 0.404 | 0.208 | 0.360 | 0.356 | 0.228 | 0.152 | 0.030 | 0.340 |
| | PAC-Bayes Bound | 1.000 | 1.000 | 1.000 | 1.000 | 1.000 | 1.000 | 1.000 | 1.000 |
| | Upper Bound | 2.562 | 2.295 | 2.269 | 2.952 | 2.253 | 2.558 | 2.567 | 2.651 |
| Layer-wise Ada + Bound | Test Error | 0.431 | 0.471 | 0.404 | 0.653 | 0.384 | 0.573 | 0.343 | 0.583 |
| | Train Error | 0.444 | 0.528 | 0.490 | 0.638 | 0.368 | 0.556 | 0.328 | 0.458 |
| | PAC-Bayes Bound | 0.645 | 0.744 | 0.696 | 0.819 | 0.580 | 0.754 | 0.532 | 0.663 |
| | Upper Bound | 0.649 | 0.760 | 0.704 | 0.853 | 0.581 | 0.771 | 0.534 | 0.668 |

categorical $P$ over $M$ models and an arbitrary categorical $Q$, parameterised by $\boldsymbol{q}$, is given by

$$\mathrm{KL}((\boldsymbol{q})\|U([M])) = \sum_{i=1}^{M} q_i \ln q_i + \ln M, \qquad (12)$$

where $[M]$ denotes the set of natural numbers up to $M$. In the case where all of the mass in $Q$ is assigned to a single model, index by $i$, the probability mass can be said to follow a Kronecker delta function, resulting in

$$\mathrm{KL}(\delta_i\|U([M])) = \ln M. \qquad (13)$$

Table 5: Performance of `mistral-7B` (various FT models: 1+3) with Task Arithmetic Ilharco et al. (2023) on BBH benchmark. The bold faced number indicates the best PAC-Bayes bound on each dataset. The shaded rows are the datasets where the base merging schemes failed to achieve meaningful learning.

| Task | Zero-Shot | | | | Task-Arithmetic | | | |
|---|---|---|---|---|---|---|---|---|
| | Test Err. | Train Err. | PB Bound | Upper Bound | Test Err. | Train Err. | PB Bound | Upper Bound |
| Logical Deduction (5 objects) | 0.933 | 0.960 | 1.000 | 1.165 | 0.703 | 0.738 | **0.887** | 0.944 |
| Hyperbaton | 0.260 | 0.210 | 0.402 | 0.415 | 0.239 | 0.172 | **0.357** | 0.377 |
| Snarks | 0.641 | 0.660 | 0.829 | 0.865 | 0.392 | 0.358 | **0.603** | 0.605 |
| Boolean Expressions | 0.247 | 0.190 | 0.379 | 0.395 | 0.247 | 0.180 | **0.368** | 0.386 |
| Tracking Shuffled Objects (7 objects) | 0.987 | 0.930 | **1.000** | 1.135 | 0.888 | 0.874 | 1.000 | 1.079 |
| Geometric Shapes | 0.920 | 0.880 | 1.000 | 1.085 | 0.757 | 0.744 | **0.897** | 0.961 |
| Web of Lies | 0.993 | 0.950 | **1.000** | 1.155 | 0.964 | 0.956 | 1.000 | 1.170 |
| Tracking Shuffled Objects (5 objects) | 0.893 | 0.960 | **1.000** | 1.165 | 0.817 | 0.838 | 1.000 | 1.049 |
| Disambiguation QA | 0.847 | 0.850 | **1.000** | 1.055 | 0.896 | 0.834 | 1.000 | 1.048 |
| Causal Judgement | 0.943 | 0.920 | 1.000 | 1.125 | 0.451 | 0.412 | **0.655** | 0.659 |
| Object Counting | 0.567 | 0.590 | **0.774** | 0.795 | 0.612 | 0.630 | 0.816 | 0.848 |
| Word Sorting | 0.940 | 0.940 | **1.000** | 1.145 | 0.939 | 0.946 | 1.000 | 1.164 |
| Multistep Arithmetic (2) | 0.993 | 1.000 | **1.000** | 1.205 | 0.996 | 0.998 | 1.000 | 1.216 |
| Reasoning About Colored Objects | 0.907 | 0.870 | 1.000 | 1.075 | 0.541 | 0.538 | **0.730** | 0.743 |
| Formal Fallacies | 1.000 | 1.000 | 1.000 | 1.205 | 0.433 | 0.476 | **0.678** | 0.684 |
| Tracking Shuffled Objects (3 objects) | 0.927 | 0.870 | 1.000 | 1.075 | 0.747 | 0.672 | **0.840** | 0.879 |
| Temporal Sequences | 1.000 | 1.000 | **1.000** | 1.205 | 0.993 | 0.998 | 1.000 | 1.207 |
| Movie Recommendation | 0.560 | 0.490 | 0.687 | 0.695 | 0.424 | 0.334 | **0.543** | 0.544 |
| Dyck Languages | 1.000 | 0.990 | **1.000** | 1.195 | 0.972 | 0.986 | 1.000 | 1.192 |
| Ruin Names | 0.940 | 0.990 | 1.000 | 1.195 | 0.749 | 0.722 | **0.902** | 0.976 |
| Navigate | 0.760 | 0.810 | 1.000 | 1.015 | 0.492 | 0.360 | **0.584** | 0.585 |
| Salient Translation Error Detection | 1.000 | 1.000 | **1.000** | 1.205 | 0.805 | 0.838 | 1.000 | 1.092 |
| Penguins in a Table | 0.891 | 0.840 | **1.000** | 1.045 | 0.878 | 0.822 | 1.000 | 1.036 |
| Date Understanding | 0.547 | 0.530 | **0.723** | 0.735 | 0.561 | 0.534 | 0.737 | 0.752 |
| Logical Deduction (3 objects) | 0.973 | 0.960 | 1.000 | 1.165 | 0.705 | 0.700 | **0.860** | 0.906 |
| Logical Deduction (7 objects) | 0.960 | 0.950 | 1.000 | 1.155 | 0.840 | 0.784 | **0.918** | 0.991 |
| Sports Understanding | 0.233 | 0.270 | 0.470 | 0.475 | 0.200 | 0.230 | **0.429** | 0.439 |

# C ADDITIONAL EXPERIMENTAL RESULTS

## C.1 RESULTS WITH CLIP-ViT-B/32

We provide full results with CLIP-ViT-B/32 (Dosovitskiy, 2020; Radford et al., 2021) on 8 vision datasets: Stanford Cars Krause et al. (2013), EuroSAT Helber et al. (2019), RESISC45 Cheng et al. (2017), GTSRB Stallkamp et al. (2012), SUN397 Xiao et al. (2010), SVHN Netzer et al. (2011), MNIST LeCun et al. (1998), and DTD Cimpoi et al. (2014), in Table 4. While the most of the algorithms are off-the-shelf non-vacuous, Layer-wise Adamerging (Yang et al., 2024) shows vacuous bounds due to its large number of parameters. However, one can easily achieve non-vacuous bounds with simple tweak on learning objectives to PAC-Bayes bound optimisation objective. Note that the cases where the improvements on PAC-Bayes certificates become marginal are the cases that actual learning by merging scheme is marginal.

## C.2 RESULTS WITH MISTRAL-7B AND LLAMA3-8B-INSTRUCTION

We provide full results with Mistral-7B (Jiang et al., 2023) and LLama3-8B-Instruction on BBH (Suzgun et al., 2022) benchmark in Table 5 and Table 6. We used 3 fintuned models for Llama3-8B-Instruct: Undi95-Meta-Llama-3.1-8B-Instruct-OAS (Met, a), LlamaFactoryAI-Llama-3.1-8B-Instruct-cv-job-description-matching (Lla), kangqi-ni-Llama-3.1-8B-Instruct-bio-tutor-sft (Kan). Notably, Task Arithmetic Ilharco et al. (2023) significantly improves the performances on the large set of tasks, both on test accuracy and PAC-Bayes bounds sense, while the baseline model tends to give poor performances and vacuous bounds. While task arithmetic gives vacuous bounds or looser bounds in some tasks, those are the tasks that Task Arithmetic Ilharco et al. (2023) fails to adapt using given data points.

Table 6: Performance of `LLama3-8B-Instruction` (various FT models: 1+3) with Task Arithmetic Ilharco et al. (2023) on BBH benchmark. The bold faced number indicates the best PAC-Bayes bound on each dataset. The shaded rows are the datasets where Task Arithmetic failed to improve over Zero-Shot.

| Task | Zero-Shot | | | | Task-Arithmetic | | | |
|------|-----------|---|---|---|-----------------|---|---|---|
| | Test Err. | Train Err. | PB Bound | Upper Bound | Test Err. | Train Err. | PB Bound | Upper Bound |
| Logical Deduction (5 objects) | 0.767 | 0.800 | **1.000** | 1.005 | 0.809 | 0.820 | **1.000** | 1.026 |
| Hyperbaton | 0.653 | 0.620 | 0.798 | 0.825 | 0.559 | 0.610 | **0.792** | 0.817 |
| Snarks | 0.333 | 0.370 | **0.574** | 0.575 | 0.418 | 0.430 | 0.638 | 0.641 |
| Boolean Expressions | 0.393 | 0.350 | **0.554** | 0.555 | 0.363 | 0.346 | 0.556 | 0.557 |
| Tracking Shuffled Objects (7 objects) | 0.913 | 0.870 | **1.000** | 1.075 | 0.895 | 0.924 | **1.000** | 1.138 |
| Geometric Shapes | 0.800 | 0.840 | **1.000** | 1.045 | 0.803 | 0.806 | **1.000** | 1.015 |
| Web of Lies | 1.000 | 1.000 | **1.000** | 1.205 | 0.999 | 1.000 | **1.000** | 1.247 |
| Tracking Shuffled Objects (5 objects) | 0.807 | 0.880 | **1.000** | 1.085 | 0.832 | 0.856 | **1.000** | 1.070 |
| Disambiguation QA | 0.760 | 0.670 | 0.837 | 0.875 | 0.752 | 0.652 | **0.827** | 0.862 |
| Causal Judgement | 0.621 | 0.590 | 0.774 | 0.795 | 0.508 | 0.552 | **0.767** | 0.787 |
| Object Counting | 0.960 | 0.940 | **1.000** | 1.145 | 0.912 | 0.918 | **1.000** | 1.132 |
| Word Sorting | 0.953 | 0.950 | **1.000** | 1.155 | 0.969 | 0.974 | **1.000** | 1.185 |
| Multistep Arithmetic (2) | 1.000 | 1.000 | **1.000** | 1.205 | 0.997 | 0.998 | **1.000** | 1.216 |
| Reasoning About Colored Objects | 0.427 | 0.480 | **0.678** | 0.685 | 0.503 | 0.488 | 0.686 | 0.694 |
| Formal Fallacies | 0.867 | 0.930 | **1.000** | 1.135 | 0.905 | 0.916 | **1.000** | 1.127 |
| Tracking Shuffled Objects (3 objects) | 0.727 | 0.800 | 1.000 | 1.005 | 0.733 | 0.744 | **0.891** | 0.950 |
| Temporal Sequences | 0.480 | 0.520 | 0.714 | 0.725 | 0.405 | 0.342 | **0.561** | 0.563 |
| Movie Recommendation | 0.653 | 0.680 | 0.845 | 0.885 | 0.671 | 0.660 | **0.832** | 0.869 |
| Dyck Languages | 0.947 | 0.980 | **1.000** | 1.185 | 0.956 | 0.960 | **1.000** | 1.174 |
| Ruin Names | 0.420 | 0.450 | 0.651 | 0.655 | 0.408 | 0.364 | **0.572** | 0.573 |
| Navigate | 0.833 | 0.870 | 1.000 | 1.075 | 0.548 | 0.482 | **0.712** | 0.723 |
| Salient Translation Error Detection | 0.673 | 0.670 | **0.837** | 0.875 | 0.673 | 0.692 | 0.860 | 0.906 |
| Penguins in a Table | 0.652 | 0.760 | **0.901** | 0.965 | 0.770 | 0.756 | 0.902 | 0.967 |
| Date Understanding | 0.753 | 0.720 | **0.874** | 0.925 | 0.745 | 0.744 | 0.896 | 0.958 |
| Logical Deduction (3 objects) | 0.720 | 0.780 | 0.914 | 0.985 | 0.733 | 0.726 | **0.884** | 0.940 |
| Logical Deduction (7 objects) | 0.800 | 0.830 | **1.000** | 1.035 | 0.833 | 0.814 | **1.000** | 1.028 |
| Sports Understanding | 0.433 | 0.470 | 0.669 | 0.675 | 0.307 | 0.348 | **0.572** | 0.573 |

Table 7: Layer-wise Adamerging Yang et al. (2024) + Bound with varying diagonal variances (0.025, 0.05, 0.1, 0.2) with Clip-ViT-B/32. across 8 image classification datasets.

| Dataset | Var=0.025 | | | Var=0.05 | | | Var=0.1 | | | Var=0.2 | | |
|---------|-----------|---|---|----------|---|---|---------|---|---|---------|---|---|
| | Test | Train | Bound | Test | Train | Bound | Test | Train | Bound | Test | Train | Bound |
| **cars** | 0.430 | 0.444 | **0.645** | 0.431 | 0.444 | **0.645** | 0.435 | 0.446 | 0.651 | 0.453 | 0.470 | 0.670 |
| **eurosat** | 0.482 | 0.516 | 0.716 | 0.471 | 0.528 | 0.744 | 0.469 | 0.510 | 0.715 | 0.478 | 0.502 | **0.712** |
| **resisc45** | 0.407 | 0.390 | 0.595 | 0.404 | 0.490 | 0.696 | 0.409 | 0.386 | **0.593** | 0.422 | 0.404 | 0.613 |
| **gtsrb** | 0.654 | 0.706 | 0.867 | 0.653 | 0.638 | **0.819** | 0.652 | 0.700 | 0.866 | 0.662 | 0.710 | 0.872 |
| **sun397** | 0.383 | 0.294 | **0.495** | 0.384 | 0.368 | 0.580 | 0.387 | 0.316 | 0.527 | 0.399 | 0.334 | 0.549 |
| **svhn** | 0.571 | 0.488 | **0.695** | 0.573 | 0.556 | 0.754 | 0.577 | 0.498 | 0.719 | 0.606 | 0.522 | 0.757 |
| **mnist** | 0.341 | 0.274 | **0.483** | 0.343 | 0.328 | 0.532 | 0.351 | 0.290 | 0.494 | 0.374 | 0.330 | 0.537 |
| **dtd** | 0.579 | 0.512 | 0.707 | 0.583 | 0.458 | **0.663** | 0.585 | 0.518 | 0.715 | 0.592 | 0.538 | 0.730 |

## C.3 HYPER-PARAMETER SENSITIVITY ANALYSIS

Table 7 illustrates the effects of prior/posterior variance on Test accuracy, Train accuracy and PAC-Bayes bound. It clearly shows that the proposed PAC-Bayesian Model Merging is robust to the selection of its prior/posterior variance.

## C.4 CHANGES ON METRICS AT DIFFERENT TRAINING STEPS

We provide the results of each metrics including Train/Test Accuracy, PAC-Bayes Bound, PAC-Bayes Upper Bound and KL-Divergence at different training steps. Figure 5 shows the change of each metric on eurosat with Clip-ViT-B/32 using Layerwise Adamerging. As learning proceeds, since the posterior deviates from the prior, the KL-Divergence term might increase. However, the total PAC-Bayes bound can still get tighter if the training error simultaneously decreases. Our bound optimisation strategy automatically manages this tradeoff. Also, the proposed Data-Dependent

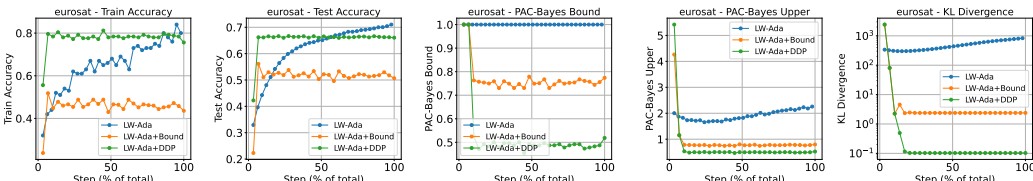

Figure 5: Change of Metrics (Train Accuracy, Test Accuracy, PAC-Bayes Bound, PAC-Bayes Upper Bound and KL-Divergence) as the training proceeds on eurosat with Clip-ViT-B/32. LW-Ada refers to Layerwise Adamerging, LW-Ada+Bound refers to directly optimising PAC-Bayes bound with Layerwise Adamerging and LW-Ada+DDP refers to directly optimising PAC-Bayes bound using Data-dependent Prior (DDP) with Layerwise Adamerging.

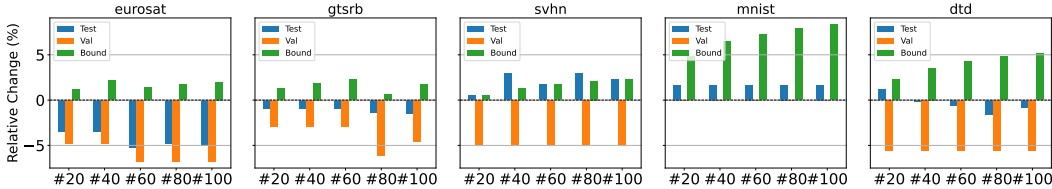

Figure 6: Relative change in Test error, Validation error, and PAC-Bayes bound for discrete compared to the continuous hypothesis class, across 5 CLIP-ViT-B/32 image classification datasets. Each subplot corresponds to a dataset, and each bar group represents a different number of discretised hypothesis classes (20, 40, 60, 80, 100). The PAC-Bayes bound exhibits a marginal increase as the number of discretised classes grows.

Prior (DDP) approach makes it possible to find a good posterior near the prior by making prior contains rich information, which results in high Train/Test Accuracy with Low KL-Divergence.

### C.4.1 CERTIFICATES FOR FINITE HYPOTHESIS CLASSES

The results so far focused on a randomized PAC-Bayes bound using Gaussian priors and posteriors (Eq. 11). As discussed in Section 3.1, discretization-based bounds are also possible (Eq. 13). These might be preferred in some applications, because they avoid the need to use a more costly randomized classifier during inference. It turns out that our insight about the certifiability of model mergers is quite general and also applies when merging algorithms are certified in this way. To illustrate this, we use Task Arithmetic Ilharco et al. (2023), and assume a finite hypothesis class uniformly distributed in $[0, 1]$, over the Task Arithmetic parameters. The results in Figure 6 show that one can achieve reasonable non-vacuous bounds even with the deterministic model, although the bounds become looser than randomised model in Section 5.2.

## D DETAILED PROCEDURE FOR PAC-BAYESIAN MODEL MERGING

---

**Algorithm 1** Off-the-shelf model merging with PAC-Bayes reinterpretation

---

**Require:** Pretrained models $\{f_1, \ldots, f_K\}$, merging algorithm $\mathcal{M}$ with parameters $\theta \in \mathbb{R}^d$, training set $S = \{(x_i, y_i)\}_{i=1}^n$, prior variance $\sigma_p^2$, posterior variance $\sigma_q^2$, confidence level $\delta$

1: $\hat{\theta} \leftarrow \mathcal{M}(\{f_k\}_{k=1}^K, S)$               ▷ run base merging algorithm

2: $P \leftarrow \mathcal{N}(\mu_P, \sigma_p^2 I)$ with $\mu_P = (1/K, \ldots, 1/K)$

3: $Q \leftarrow \mathcal{N}(\mu_Q, \sigma_q^2 I)$ with $\mu_Q = \hat{\theta}$

4: Estimate empirical risk of $Q$ via Monte Carlo:
         Sample $\{\phi_t\}_{t=1}^T \sim Q$
         For each $\phi_t$, build merged model $f_{\phi_t}$
         $\hat{L}(Q) \approx \frac{1}{T} \sum_{t=1}^T \frac{1}{n} \sum_{i=1}^n \ell(f_{\phi_t}(x_i), y_i)$

5: Compute $\mathrm{KL}(Q\|P)$ using the closed form for diagonal Gaussians

6: Find $C$ such that $\mathrm{kl}(\hat{L}(Q)\|C) = (\mathrm{KL}(Q\|P) + \log(n/\delta))/(n-1)$ using Theorem 2

7: **return** merged parameters $\hat{\theta}$ and certificate $C$

---

---

**Algorithm 2** PAC-Bayes bound-optimised model merging

---

**Require:** Pretrained models $\{f_1, \ldots, f_K\}$, training set $S = \{(x_i, y_i)\}_{i=1}^n$, prior $P = \mathcal{N}(\mu_P, \sigma_p^2 I)$ with $\mu_P = (1/K, \ldots, 1/K)$, posterior family $Q(\theta) = \mathcal{N}(\theta, \sigma_q^2 I)$, gradient-free optimiser (e.g. CMA-ES), iterations $T$, confidence level $\delta$
1: $\theta_0 \leftarrow \mu_P$
2: **for** $t = 0$ to $T - 1$ **do**
3:     Sample candidate parameters $\{\theta_t^{(j)}\}_j$ from the optimiser
4:     **for** each candidate $\theta_t^{(j)}$ **do**
5:         $Q_j \leftarrow \mathcal{N}(\theta_t^{(j)}, \sigma_q^2 I)$
6:         Estimate $\hat{L}(Q_j)$ on $S$ via Monte Carlo sampling from $Q_j$
7:         Compute $\mathrm{KL}(Q_j \| P)$
8:         Find $C$ such that $\mathrm{kl}(\hat{L}(Q_j) \| C) = \big(\mathrm{KL}(Q_j \| P) + \log(n/\delta)\big)/(n - 1)$
9:         $B(\theta_t^{(j)}) \leftarrow C$
10:     **end for**
11:     Feed $\{B(\theta_t^{(j)})\}_j$ to the optimiser as objective values
12:     Update optimiser state and set $\theta_{t+1}$ to the best candidate so far
13: **end for**
14: Let $\theta^\star$ be the parameter with the smallest bound $B(\theta)$
15: **return** posterior $Q^\star = \mathcal{N}(\theta^\star, \sigma_q^2 I)$ and certificate $B(\theta^\star)$

---

**Algorithm 3** Constructing a data-dependent prior (DDP) for model merging

---

**Require:** Training set $S = \{(x_i, y_i)\}_{i=1}^n$, pretrained models $\{f_1, \ldots, f_K\}$, prior variance $\sigma_p^2$, posterior variance $\sigma_q^2$, confidence level $\delta$
1: Split $S$ into disjoint subsets: $S_{\text{prior}}$ and $S_{\text{support}}$
2: **Prior construction:**
    Apply Algorithm 2 (or Algorithm 1) on $S_{\text{prior}}$ to obtain $\theta_{\text{prior}}$
    $P_{\text{DDP}} \leftarrow \mathcal{N}(\theta_{\text{prior}}, \sigma_p^2 I)$
3: **Certified learning on support set:**
    Run Algorithm 2 on $S_{\text{support}}$ using $P_{\text{DDP}}$ as prior
    Obtain $\theta^\star$ and $Q^\star = \mathcal{N}(\theta^\star, \sigma_q^2 I)$ with bound $B_{\text{DDP}}$
4: **return** prior $P_{\text{DDP}}$, posterior $Q^\star$, and certificate $B_{\text{DDP}}$

---

