# OpenReview forum: "Model Merging is Secretly Certifiable: Non-Vacuous Generalisation Bounds for Low-Shot Learning"
_ICLR.cc/2026/Conference — Submitted to ICLR 2026_

### Official Review · Reviewer_xkfc · 2025-10-29

**Soundness:** 3
**Presentation:** 2
**Contribution:** 3
**Rating:** 6
**Confidence:** 3

**Summary:**

This work presents a novel approach to provide non-vacuous generalization guarantees for large neural networks trained on low-shot data. The authors make an insightful observation: by reformulating the learning problem from "fine-tuning the entire model" to "learning how to merge the weights of multiple pre-trained models", they can leverage the PAC-Bayes framework so that the generalization bound depends only on the number of merging parameters, rather than the scale of the underlying models. This is an interesting and meaningful work, as it provides a new perspective on how to theoretically analyze and improve the generalization of large models in low-shot regimes.

**Strengths:**

1. The main strength of this paper lies in its being the first to provide non-vacuous generalization guarantees for modern large-scale models under extremely low-shot settings.
2. By shifting the focus from fine-tuning to model merging, the authors successfully decouple the difficulty of providing generalization guarantees. The generalization gap no longer depends on the enormous scale of the underlying models, but only on the relatively small number of parameters involved in the merging process.
3. The authors further demonstrate that many existing off-the-shelf model merging algorithms can be certified to provide non-vacuous generalization guarantees with only minor modifications.

**Weaknesses:**

1. The paper does not report the runtime or computational overhead of this "certified optimization", which raises concerns about its practical feasibility.
2. If a suitable pool of source models cannot be found, or if the source models themselves are of low quality, the merging algorithm is likely to fail—rendering the proposed "certification" meaningless in practice.
3. In truly extreme low-shot scenarios (e.g., 10-shot), further splitting the data could severely compromise training stability and the model’s final performance.

**Questions:**

1. If the quality of the source model pool deteriorates—for example, if the models are unrelated to the downstream task, or if the models are too similar or too diverse—how would the PAC-Bayes guarantees in this paper be affected?
2. Compared to the original objective, how much additional training time or computational resources (e.g., GPU hours) does Bound Optimization require?
3. Is the data-splitting strategy still feasible in extremely low-data scenarios (e.g., (n=20) or (n=10))?

---

> ### Author Response · Authors · 2025-11-20
> **Answers to Weakness/Question 1 and 2**
>
> We thank the reviewer for the time and effort spent carefully evaluating our work. The following responses address the concerns raised. Please let us know if any further clarification is needed.
>
> ***W1.The paper does not report the runtime or computational overhead of this "certified optimization", which raises concerns about its practical feasibility.***
>
> TLDR: The bound computation adds negligible overhead compared to the baseline merging procedures.
>
> The optimisation minimises the training loss plus a PAC-Bayesian term, but the only additional quantity that must be computed is the closed-form $\mathrm{KL}(P\Vert Q)$ between diagonal Gaussians, which is computationally trivial.
>
> The empirical risk term $\hat{L}(Q)$ requires Monte-Carlo approximation, but using a single posterior sample ($n=1$) is a standard and widely adopted practice (e.g., Dziugaite & Roy, 2017) and incurs the same computational cost as evaluating the baseline merging method once. With CLIP-ViT-B/32, CMA-ES, and Task-wise AdaMerging, both the baseline merge and the bound-optimisation procedure have identical wall-clock cost of 1.53s/epoch when using a single posterior sample.
>
> To provide a demonstration that this approximation does not affect the final certification quality, we compare results with $n=1$ and $n=5$ posterior samples. The two settings yield nearly identical test performance and PAC-Bayes bounds (see Table below), indicating that $n=1$ is sufficient in practice.
> | Dataset | Test Err (n=5) | Train Err (n=5) | PB Bound (n=5) | Upper (n=5) | Test Err (n=1) | Val Err (n=1) | PB Bound (n=1) | Upper (n=1) |
> |---------|----------------|------------------|----------------|-------------|----------------|----------------|----------------|--------------|
> | eurosat | 0.341 | 0.376 | 0.613 | 0.615 | 0.3988 | 0.3780 | 0.6099 | 0.6119 |
> | gtsrb   | 0.645 | 0.570 | 0.776 | 0.798 | 0.6538 | 0.5620 | 0.7779 | 0.8015 |
> | svhn    | 0.340 | 0.300 | 0.541 | 0.544 | 0.3396 | 0.3060 | 0.5577 | 0.5599 |
> | mnist   | 0.149 | 0.130 | 0.345 | 0.375 | 0.1478 | 0.1380 | 0.3611 | 0.3885 |
> | dtd     | 0.568 | 0.426 | 0.638 | 0.641 | 0.5776 | 0.4520 | 0.6591 | 0.6637 |
>
>
> ***W2. If a suitable pool of source models cannot be found, or if the source models themselves are of low quality, the merging algorithm is likely to fail—rendering the proposed "certification" meaningless in practice.***
>
> ***Failing Merging:***
>
> Our goal is to certify the generalisation performance of large models in settings where the underlying learning succeeds. If the merging algorithm fails because the pool of source models is extremely poor or fundamentally misaligned with the target, then there is no meaningful predictor to certify. This limitation is not specific to our method: any guarantee (learning-theoretic or otherwise) becomes vacuous when the underlying learner cannot form a useful hypothesis.
>
> In practice, however, model merging does not require a carefully curated or highly task-aligned source pool. Akiba et al. (2025) report that merging can produce strong downstream performance even when the constituent models are trained on unrelated tasks, indicating that suitable source pools are often easy to obtain.
>
> Importantly, the certification framework we propose is independent of the specific merging algorithm. The PAC-Bayesian guarantee applies to whichever predictor the merge operator produces. If a particular merging strategy fails on certain tasks, the issue lies with the merge operator rather than with the certification framework, and more recent or robust merging algorithms can be used in its place.
>
> ***No Source Models at All:***
>
> More generally, for tackling few-shot problems, all solutions rely on some kind of prior knowledge - either human prior knowledge or a prior in the form of related tasks [Song 2023] - to constrain the problem. In the absence of any prior knowledge (literally no source models), no larger learner will succeed a low-shot regime. IE: The “no source models/tasks” concern is common to all few-shot learners, not only ours.

---

> ### Author Response · Authors · 2025-11-20
> **Answer to Weakness/Question 3**
>
> ***W3.In truly extreme low-shot scenarios (e.g., 10-shot), further splitting the data could severely compromise training stability and the model’s final performance.***
>
> TLDR: “Splitting the data” in PAC-Bayes DDP sense doesn’t degrade training in the way that it does in train/val workflows. All data still contributes.
>
> ***Compromise from data Split?***
>
> The PAC-Bayesian framework is useful for low-shot scenarios because it provides certification while still allowing the learner to fully utilise the available data. Unlike validation-based approaches that reserve a portion of the data for hold-out evaluation, PAC-Bayes uses all examples for estimating the empirical risk that appears in the bound.
>
> In the data-dependent prior (DDP) construction, the dataset is split into two disjoint parts, but both subsets still contribute to learning: one for constructing the prior and the other for training the posterior. Thus, the full dataset is used in a two-stage manner rather than being partially discarded. As shown in Figure 3, this leads to tighter bounds than the standard validation-split strategy, which effectively wastes half of the available samples.
>
> ***Is 10-shot feasible?***
>
> Nevertheless, it is important to note that there is an inherent limitation in PAC-Bayesian bounds—and in fact, in any generalisation-theoretic analysis—that is independent of model complexity. The bound contains the term  $\log(n/\delta)/{n}$ which becomes large when n is extremely small. This behaviour appears even when the effective number of learnable parameters is close to zero, implying that vacuous bounds are unavoidable in extreme few-shot scenarios such as 1–5 examples. This limitation arises from the statistical nature of the problem rather than from the DDP construction or the merging method itself.
>
> Despite this universal limitation, Figure 3 shows that our framework can still certify near–extremely few-shot learning where prior work becomes infeasible. Notably, the strongest existing results for large models (e.g., Lotfi et al., 2024) require tens of thousands of training samples, whereas our approach yields meaningful certificates with samples fewer than 100.
>
> For completeness, we now provide certification results under 10-shot setting with Taskwise Adamerging+Bound optimisation approach below:
>
> | Method                     | Metric       | eurosat | gtsrb | svhn | mnist | dtd  |
> |---------------------------|--------------|---------|-------|------|-------|------|
> | TW-Ada + Bound Opt.       | Test Error   | 0.438   | 0.651 | 0.370 | 0.167 | 0.564 |
> |                           | Train Error  | 0.560   | 0.280 | 0.240 | 0.260 | 0.020 |
> |                           | PB Bound     | 1.000   | 0.834 | 0.870 | 0.876 | 0.597 |
> |                           | Upper Bound  | 1.149   | 0.892 | 0.959 | 0.969 | 0.653 |
>
> While 10-shot is small enough to be affected by the fundamental barrier of the  $\log(n/\delta)/{n}$ term, some non-vacuous certificates are still obtained.

---

> > ### Comment · Reviewer_xkfc · 2025-11-25
> >
> > Thank you for the clear and detailed rebuttal. My earlier concerns have been satisfactorily addressed. Regarding the overall contribution and novelty, my assessment remains similar to the original one.

---

> > > ### Author Response · Authors · 2025-11-27
> > >
> > > We sincerely appreciate the reviewer’s continued positive evaluation.
> > > If the reviewer has any additional questions or requires further clarification, we would be pleased to provide it.

---

### Official Review · Reviewer_LQRu · 2025-10-31

**Soundness:** 2
**Presentation:** 2
**Contribution:** 2
**Rating:** 4
**Confidence:** 4

**Summary:**

This paper aims to make the certification of IID generalization feasible for large neural networks trained on small datasets. It establishes a principled link between model fusion methods and PAC-Bayes theory, showing that learning only a small set of fusion weights yields a low-dimensional stochastic predictor with a generalization gap independent of model size. Building on this insight, the paper presents several new findings. (i) Simple, off-the-shelf model-merging learners can already achieve non-trivial guarantees. (ii) These guarantees hold under multiple standard formulations, including Gaussian PAC-Bayes and discretization bounds. (iii) Modifying the merge objective to directly optimize the PAC-Bayes bound, instead of the training likelihood alone, enables more advanced merging methods to attain both strong test performance and tight certificates. (iv) The study provides the first demonstration of non-vacuous generalization guarantees for large models such as ViT-B and Mistral-7B in the low-shot regime with only 100 labeled examples. Together, these contributions reveal a practical path toward certifying large pre-trained systems as trustworthy without full fine-tuning, offering a promising synthesis of theory and practice.

**Strengths:**

1. The paper introduces an intellectually appealing idea by identifying a novel connection between model fusion methods and generalization certification. This perspective leads to a new form of generalization bound that unifies practical model-merging strategies with formal theoretical guarantees, offering a concise and elegant framework bridging empirical practice and generalization theory.
2. The paper presents comprehensive empirical validation. It evaluates the proposed generalization bounds across diverse datasets and large-scale vision and language models, providing strong and credible experimental support for the claimed theoretical contributions.

**Weaknesses:**

1. The exposition of the PAC-learning framework in Section 3 lacks conceptual precision. The PAC framework does not seek to bound the discrepancy between the empirical risk and the population risk for an arbitrary, fixed parameter $\theta$. For any $\theta$, that is independent of the data, the difference between $L(\theta)$ and $\hat{L}(\theta)$ converges at the standard Monte Carlo rate $O(1/n)$,taking square error loss function as an example. This is a purely sampling effect rather than a learning-theoretic phenomenon. The proper PAC formulation considers the data-dependent estimator $\theta* = \operatorname{argmin}_{\theta} \hat{L}(\theta)$ and establishes bounds on the generalization gap $L(\theta*) - \hat{L}(\theta*)$. Since $\theta^*$ depends on the training sample, this gap reflects the algorithmic dependence on data and typically converges at a slower rate than $1/n$.

2. The paper’s motivation is insufficiently articulated, leaving the relevance of its setting unclear. (i) The focus on small datasets with highly overparameterized models is theoretically questionable. In this regime, approximation error is minimal, but model complexity is excessive, necessitating strong regularization and typically leading to poor generalization. A more principled analysis would consider the classical regime where the sample size $n$ grows and model complexity is scaled accordingly, allowing examination of the optimal bias–variance trade-off and the convergence rate of the generalization error as a function of  $n$  (ii) The analysis implicitly assumes near-optimal optimization, i.e., that the obtained parameter $\theta^*$, closely matches the empirical risk minimizer. In large-scale models, however, computational constraints prevent full optimization, making such assumptions unrealistic. Under these conditions, analyzing generalization becomes less informative than studying scaling laws, which capture how performance evolves with model and data size under practical optimization dynamics.

3. The paper provides limited theoretical justification or intuitive insight into why the proposed generalization bound improves upon conventional ones. While the empirical evidence suggests tighter or non-vacuous guarantees, the manuscript does not clearly articulate the underlying mechanism or theoretical principle responsible for this improvement.

**Questions:**

See weaknesses.

---

> ### Author Response · Authors · 2025-11-20
> **Answer to Weakness 1**
>
> We thank the reviewer for the time and effort spent carefully evaluating our work. The following responses address the concerns raised. Please let us know if any further clarification is needed.
>
> **W1. The exposition of the PAC-learning framework in Section 3 lacks conceptual precision.
> The PAC framework does not seek to bound the discrepancy between the empirical risk and the population risk for an arbitrary, fixed parameter $\\theta$.
> For any $\\theta$ that is independent of the data, the difference between $L(\\theta)$ and $\\hat{L}(\\theta)$ converges at the standard Monte Carlo rate $O(1/n)$, taking square error loss as an example.
> This is a purely sampling effect rather than a learning-theoretic phenomenon.**
>
> **The proper PAC formulation considers the data-dependent estimator**
>
> $$
> \\theta^* = \\arg\\min_{\\theta} \\hat{L}(\\theta)
> $$
>
> **and establishes bounds on the generalization gap**
>
> $$
> L(\\theta^\*) - \\hat{L}(\\theta^\*).
> $$
>
> **Since $\\theta^*$ depends on the training sample, this gap reflects the algorithmic dependence on data and typically converges at a slower rate than $1/n$.**
>
>
>
>
>
> The reviewer’s comment conflates the classical PAC-learning setting with the PAC-Bayesian framework used in our paper. Classical PAC-learning is most commonly concerned with uniform convergence over a hypothesis class, and the behaviour of a fixed, data-independent hypothesis is governed simply by standard concentration inequalities. This “fixed-parameter concentration” phenomenon is not the object of PAC-learning itself, and it is unrelated to the theoretical setting (PAC-Bayesian framework) in our work.
>
> Our analysis is based on PAC-Bayesian theory, which provides generalisation bounds for any stochastic predictor (PAC-Bayes Posterior) $Q$. PAC-Bayes theorems (e.g., Langford & Seeger, 2001) hold for any prior $P$, which is independent of the data, and any posterior $Q$.  In this framework, $\mathrm{KL}(P\Vert Q)$ term acts as the complexity term that controls how far the learned posterior can deviate from the data-independent prior. This is the quantity that governs the generalisation gap in PAC-Bayesian analysis, and it is fundamentally different from the reviewer’s fixed-parameter concentration argument, which does not apply to the PAC-Bayesian setting.
>
> Importantly, PAC-Bayesian theorems already guarantee the relevant form of uniform convergence for the entire posterior family. Consequently, minimising the right-hand side of the Langford–Seeger bound (Theorem 2) corresponds exactly to minimising the certified generalisation error. The objective used in our optimisation directly follows this principle: reducing the empirical term and the KL term leads to a tighter certification, with no reliance on classical PAC assumptions.
>
> Section 3 accurately reflects the PAC-Bayesian framework, and the critique based on classical PAC-learning does not apply to our analysis.

---

> ### Author Response · Authors · 2025-11-20
> **Answer to Weakness 2**
>
> **W2. The paper’s motivation is insufficiently articulated, leaving the relevance of its setting unclear.**
>
> **(i) The focus on small datasets with highly overparameterized models is theoretically questionable.**
> **In this regime, approximation error is minimal, but model complexity is excessive, necessitating strong regularization and typically leading to poor generalization.**
>
> **A more principled analysis would consider the classical regime where the sample size $n$ grows and model complexity is scaled accordingly, allowing examination of the optimal bias–variance trade-off and the convergence rate of the generalization error as a function of $n$.**
>
> **(ii) The analysis implicitly assumes near-optimal optimization, i.e., that the obtained parameter $\theta^*$ closely matches the empirical risk minimizer.**
> **In large-scale models, however, computational constraints prevent full optimization, making such assumptions unrealistic.**
> **Under these conditions, analyzing generalization becomes less informative than studying scaling laws, which capture how performance evolves with model and data size under practical optimization dynamics.**
>
>
> The reviewer’s point in (i) misinterprets the setting we consider.
>
> As stated in L15–17, L51–53, and L108–112, the paper begins from the observation that certification in the full parameter space of large models is infeasible: the number of backbone parameters is orders of magnitude larger than the available data, making classical generalisation analyses vacuous, particularly in low-shot scenarios. For this reason, and as emphasised in L20–22, L61–65, L101–104, L119–121, and L254–255, we focus on model merging precisely because it provides an extremely compact parameterisation in which certification is tractable. All PAC-Bayesian analysis in the paper is performed in this low-dimensional merging space, not in the overparameterised backbone. The reviewer’s concern therefore addresses a regime different from the one we consider. In fact the comment references a regime that our paper deliberately avoids and explicitly motivates moving away from.
>
> The reviewer’s statement in (ii) reflects a misunderstanding of both our optimisation setting and the requirements of the PAC-Bayesian framework.
>
> Our analysis does not assume “near-optimal optimisation” for the learned predictor. PAC-Bayesian bounds hold for any posterior distribution $Q$, regardless of how well it minimises the empirical loss, as explicitly stated in Theorem 1 and 2. Moreover, the notion of an empirical-risk minimiser does not play any role in PAC-Bayes, because the learned object is a distribution $Q$ over predictors rather than a single parameter estimate. Consequently, terms such as “near-ERM” have no meaningful interpretation in the PAC-Bayesian setting, and our optimisation procedure is not designed to approximate an ERM solution.
>
> Bringing together the misunderstanding (i) and (ii), the reviewer’s additional claim that large-scale models cannot be fully optimised and thus “near-optimality assumptions are unrealistic” is misplaced. Our optimisation takes place entirely in the low-dimensional merging parameter space, not in the full parameter space of the backbone model. Computational constraints of large-scale backbone optimisation have no relevance to our setting, since the backbone is frozen and only a compact set of merge coefficients is updated. Moreover, no such “near-optimality assumption” is stated, required, or implied anywhere in the paper; PAC-Bayesian bounds hold for any posterior $Q$, independent of its optimisation accuracy. The premise of the reviewer’s argument does not correspond to the theoretical assumptions, optimisation procedure, or learning problem considered in the paper.
>
> The reference to scaling laws is not relevant to the problem considered in the paper. Scaling laws describe empirical behaviour when both model size and dataset size grow together under full optimisation of the backbone. They do not provide generalisation error certificates and have no applicability in the low-shot setting. Our contribution relies on a different and explicit insight: by moving learning into the compact model-merging parameter space, certification becomes independent of the size of the underlying backbone model. In this regime, the backbone remains frozen and only a small number of merging coefficients are learned, making non-vacuous certification feasible even for very large models, even with low-shot scenarios. Scaling-law analyses cannot substitute for, or invalidate, this certification perspective, since they address a fundamentally different empirical phenomenon.

---

> ### Author Response · Authors · 2025-11-20
> **Answer to Weakness 3**
>
> ***W3.The paper provides limited theoretical justification or intuitive insight into why the proposed generalization bound improves upon conventional ones. While the empirical evidence suggests tighter or non-vacuous guarantees, the manuscript does not clearly articulate the underlying mechanism or theoretical principle responsible for this improvement.***
>
> The reviewer’s comment overlooks the central theoretical mechanism repeatedly stated throughout the manuscript. As explained in L20–22, L61–65, L101–104, L119–121, and L254–255, the reason our model-merging-based certification approach yields tighter and non-vacuous guarantees is because model merging fundamentally reduces the effective hypothesis dimension. PAC-Bayesian complexity depends on the $\mathrm{KL}(P\Vert Q)$ term, whose scale is determined by the dimensionality of the parameter space on which learning takes place. In conventional finetuning, the parameter space contains millions (or even billions) of backbone parameters, making the KL term extremely large and the bound necessarily vacuous in low-data regimes. Even parameter-efficient tuning approaches such as SubLoRA (Lotfi et al., 2024) require tens of thousands of datapoints to obtain non-vacuous PAC-Bayesian bounds, precisely because their effective parameter dimension remains too large. In contrast, model merging restricts learning to an extremely compact parameterisation—often just tens of parameters—so the complexity term scales with the number of merged models rather than the size of the backbone. This explains why non-vacuous certification becomes feasible even for very large models and very small number of data. The manuscript states this mechanism clearly and repeatedly; the improvement is a direct consequence of applying PAC-Bayesian analysis to a compact merging space, not an unexplained empirical effect.

---

### Official Review · Reviewer_VEYW · 2025-11-01

**Soundness:** 3
**Presentation:** 3
**Contribution:** 2
**Rating:** 4
**Confidence:** 3

**Summary:**

The paper presents an interesting theoretical and empirical finding: model merging, e.g. combining pretrained models via learned linear interpolation, can yield non-vacuous generalization bounds under the PAC-Bayes framework, even in few-shot settings and for large models like ViT-B and Mistral-7B.
The key insight is that the learnable parameter space in model merging depends on the number of source models, not their size, making certification feasible. The authors (i) reinterpret existing merging algorithms within PAC-Bayesian theory, (ii) propose minor modifications such as bound-aware objectives and data-dependent priors, and (iii) empirically demonstrate certifiable low-shot learning on image and text tasks.

**Strengths:**

- The paper establishes an elegant bridge between model merging and PAC-Bayesian certification, a previously unexplored direction.
- It achieves non-vacuous generalization guarantees with large pretrained networks (ViT-B, Mistral-7B) using as few as 100 examples, which is unprecedented in the certification literature.
- Requires only small modifications to existing merging pipelines; bound-aware optimization and Gaussian priors are straightforward.
- Experiments cover both vision and language domains, multiple merging algorithms, and detailed ablations (bound optimization, data-dependent priors, scaling with data).
- The work suggests a feasible path for certifying modern AI systems in low-data, high-stakes contexts.

**Weaknesses:**

- While well-motivated, the paper does not provide new generalization bounds - most results hinge on reinterpreting existing theory.
- Certification results are mostly numerical bounds rather than guarantees under formal verification (e.g., robustness or adversarial guarantees).
- Some methodological explanations (e.g., computation of the PAC-Bayes term, choice of priors, gradient-free optimization) could be made more rigorous and reproducible.
- Results exclude tasks where merging fails; it would be useful to quantify how often this happens.
- Bound computations may become costly for real-world high-dimensional merges; the paper could better discuss computational complexity.

**Questions:**

- How sensitive are the certification results to the choice of variance in the Gaussian prior/posterior (e.g., $\lambda_1, \lambda_2$)?
- Does the bound remain non-vacuous if the number of source models (hence $\alpha$-parameters) grows beyond a few dozen?
- In what ways does data-dependent prior construction risk information leakage from the support set?
- Could this framework be adapted to nonlinear merging or adapter-based composition (e.g., LoRA fusion)?
- How would the results compare if the merging coefficients were learned jointly with fine-tuning of small subsets of parameters?

---

> ### Author Response · Authors · 2025-11-20
> **Answers to Weakness 1 and 2**
>
> We thank the reviewer for the time and effort spent carefully evaluating our work. The following responses address the concerns raised. Please let us know if any further clarification is needed.
>
> ***W1. While well-motivated, the paper does not provide new generalization bounds - most results hinge on reinterpreting existing theory.***
>
> Although one might view it as an incremental discovery by combining existing literature, finding a suitable sparse parameterisation and bound is meaningful itself. For instance, Lotfi et al 2024 tried a combination of LoRA-based parameterization with discretization bound (both essentially known techniques), but only succeeded to certify in a much higher data regime than us. We make the novel observation that pairing model merging with PAC-Bayes bounds, leads to non-vacuous bounds even for a few shot regime with large scale neural networks.
>
> Our finding has high practical impact because (1) It is widely believed that certifying few-shot learning is a difficult open problem (Shimabucoro et al 2024), making our finding significant in the wide variety of few-shot learning applications. (2) As model merging is a popular learner parameterization, our result reveals that a large family of existing implementations are likely already certifiable, with minimal - or no - changes.
>
> ***W2. Certification results are mostly numerical bounds rather than guarantees under formal verification (e.g., robustness or adversarial guarantees).***
>
> We interpret the reviewer’s comment as referring to formal verification methods commonly used in robustness verification. These methods analyse whether a neural network satisfies a logically defined property over a given input space, often through worst-case reasoning or perturbation-bounded checks. Importantly, such verification does not address distribution-level behaviour or test-time generalisation. These techniques evaluate properties on observed data (e.g, training or validation sets) or structured subsets of the input space, but do not provide guarantees about the model’s expected performance under the data distribution [R1].
>
> Our work focuses instead on generalisation certification, which goes beyond the training set to provide guarantees on the generalisation error for unseen data. Our results are theoretical guarantees on generalisation error and should not be confused with empirical estimates. The term “numerical” in our paper refers to evaluating a closed-form PAC-Bayes bound, not to an empirical estimate.
>
> We hope the above clarifies that our contribution is important yet different to adversarial robustness guarantees. Nevertheless, although adversarial robustness is outside the main scope of the paper, the same certification we use extends to producing bounds on the adversarial generalisation error. Specifically, PAC-Bayes framework is not tied to standard supervised losses. The bound applies to any loss taking values in $[0,1]$, so we can instantiate the same certificate for adversarial risk simply by redefining the loss.
>
> For an $\\epsilon$-bounded perturbation set, define the adversarial loss:
> $$
> \\ell\_{\\mathrm{adv}}(f\_\\theta, x, y)
> = \\max\_{\\|\\delta\\| \\le \\epsilon} \\, \\ell(f\_\\theta(x + \\delta), y).
> $$
>
> The corresponding empirical and population adversarial risks are:
> $$
> \\hat{L}\_{\\mathrm{adv}}(Q)
> = \\frac{1}{n} \\sum\_{i=1}^n
> \\mathbb{E}\_{\\theta \\sim Q}\\left[\\ell\_{\\mathrm{adv}}(f\_\\theta, x\_i, y\_i)\\right],
> $$
>
> $$
> L\_{\\mathrm{adv}}(Q)
> = \\mathbb{E}\_{(x,y) \\sim \\mathcal{D}}
> \\mathbb{E}\_{\\theta \\sim Q}\\left[\\ell\_{\\mathrm{adv}}(f\_\\theta, x, y)\\right].
> $$
>
> Since $\\ell\_{\\mathrm{adv}} \\in [0,1]$, the Langford--Seeger PAC-Bayes bound applies without modification:
> $$
> \\mathrm{kl}\\!\\left(\\hat{L}\_{\\mathrm{adv}}(Q) \\,\\|\\, L\_{\\mathrm{adv}}(Q)\\right)
> \\le
> \\frac{\\mathrm{KL}(Q\\|P) + \\log(n/\\delta)}{n - 1}.
> $$
>
> Thus, the same certificate holds for adversarial performance; the only practical change is that $\\hat{L}(Q)$ is computed using adversarial loss (e.g., via PGD).
>
> To demonstrate this, we include PGD-based adversarial results on MNIST using Taskwise Adamerge:
> | Method              | Metric      | PGD-MNIST |
> |---------------------|------------|-----------|
> | LW-Ada + DDP        | Test Error | 0.249     |
> |                     | Train Error| 0.176     |
> |                     | PB Bound   | 0.438     |
> |                     | Upper Bound| 0.454     |
>
> For adversarial attack, we utilise PGD attack with step number 7, step size 2/255, threshold 8/255.
>
> Note that these guarantees are weaker than those of the formal verification in the sense that they assume a particular attack strategy (PGD in this case). However, they are stronger in the sense that they are guarantees on the adversarial generalisation error not the robustness on the observed samples.
>
> [R1] Huasonng et al, Adversarial Robustness of Deep Neural Networks: A survey from a Formal Verification Perspective, arxiv:2206.12227, 2022

---

> ### Author Response · Authors · 2025-11-20
> **Answers to Weakness 3,4 and 5**
>
> ***W3. Some methodological explanations (e.g., computation of the PAC-Bayes term, choice of priors, gradient-free optimization) could be made more rigorous and reproducible.***
>
> Most implementation details, including the hyperparameter settings and the optimisation procedure, are already provided in Section 5.1 and Appendix A. We also include the code in the supplementary material to ensure reproducibility. While methodological details are explained in Section 4 and Appendix B, we now additionally include three short algorithm summaries in the Appendix D: (i) an off-the-shelf merging baseline with a PAC-Bayesian interpretation, (ii) the bound-optimisation variant, and (iii) the data-dependent prior construction for better understanding. These give a step-by-step description of how the prior and posterior are instantiated, how the PAC-Bayes term is computed, and how the optimisation is carried out.
>
> ***W4.Results exclude tasks where merging fails; it would be useful to quantify how often this happens.***
>
> We include all results, including failure cases of model merging, in Appendix C.
>  Quantifying “how often merging fails” in a universal sense is difficult, as the outcome strongly depends on the target data distribution, the choice of merging algorithm, the number of available samples, and the number or diversity of source models.
>
> Importantly, the proposed certification procedure is orthogonal to the specific merging algorithm. When merging does fail, the issue stems from the merge operator rather than the certification framework, and such cases can be addressed using newer or more robust families of merging algorithms.
>
> ***W5.Bound computations may become costly for real-world high-dimensional merges; the paper could better discuss computational complexity.***
>
> The bound computation imposes no measurable computational overhead over the base learner.
>
> Specifically, the optimisation minimises the training loss plus a PAC-Bayesian term, but the only additional quantity that must be computed is the closed-form $\mathrm{KL}(P\Vert Q)$ between diagonal Gaussians, which is computationally trivial.
>
> The empirical risk term $\hat{L}(Q)$ requires Monte-Carlo approximation, but using a single posterior sample ($n=1$) is a standard and widely adopted practice (e.g., Dziugaite & Roy, 2017) and incurs the same computational cost as evaluating the baseline merging method once. With CLIP-ViT-B/32, CMA-ES, and Task-wise AdaMerging, both the baseline merge and the bound-optimisation procedure have identical wall-clock cost of 1.53s/epoch when using a single posterior sample.
>
> To confirm that this approximation does not affect the final certification quality, we compare results with $n=1$ and $n=5$ posterior samples. The two settings yield nearly identical test performance and PAC-Bayes bounds (see Table below), indicating that $n=1$ is sufficient in practice.
>
> | Dataset | Test Err (n=5) | Train Err (n=5) | PB Bound (n=5) | Upper (n=5) | Test Err (n=1) | Val Err (n=1) | PB Bound (n=1) | Upper (n=1) |
> |---------|----------------|------------------|----------------|-------------|----------------|----------------|----------------|--------------|
> | eurosat | 0.341 | 0.376 | 0.613 | 0.615 | 0.3988 | 0.3780 | 0.6099 | 0.6119 |
> | gtsrb   | 0.645 | 0.570 | 0.776 | 0.798 | 0.6538 | 0.5620 | 0.7779 | 0.8015 |
> | svhn    | 0.340 | 0.300 | 0.541 | 0.544 | 0.3396 | 0.3060 | 0.5577 | 0.5599 |
> | mnist   | 0.149 | 0.130 | 0.345 | 0.375 | 0.1478 | 0.1380 | 0.3611 | 0.3885 |
> | dtd     | 0.568 | 0.426 | 0.638 | 0.641 | 0.5776 | 0.4520 | 0.6591 | 0.6637 |

---

> ### Author Response · Authors · 2025-11-20
> **Answers to Question 1, 2 and 3**
>
> ***Q1. How sensitive are the certification results to the choice of variance in the Gaussian prior/posterior?***
>
> The sensitivity analysis with respect to the prior/posterior variance is presented in Table 7 and Appendix C.3. As shown there, the certification results are stable across diverse variance settings.
>
> ***Q2. Does the bound remain non-vacuous if the number of source models (hence -parameters) grows beyond a few dozen?***
>
> In the Layerwise AdaMerge setting with CLIP-ViT/B/32, the effective number of merge parameters scales with the number of source models (24 parameters per model), reaching 168 parameters when using seven source models. As shown in Tables 1 and 2, the proposed certification framework still yields non-vacuous bounds at this scale, particularly when using the DDP construction. This indicates that the method remains stable even when the dimensionality of the merge parameters exceeds a hundred.
>
> ***Q3. In what ways does data-dependent prior construction risk information leakage from the support set?***
>
> There is no support set leakage.
>
> PAC-Bayes guarantees require the prior to be independent of the data used for posterior learning. A data-dependent prior remains valid as long as the data used to construct the prior and the data used to train the posterior are disjoint. Following the treatment in Pérez-Ortiz et al. (2022), we split the training (support) set into two independent subsets: one for constructing the prior and one for optimising the posterior. Because the posterior support set is never used when forming the prior, there is no information leakage between the two stages, and the standard PAC-Bayes reasoning continues to apply.

---

> ### Author Response · Authors · 2025-11-20
> **Answers to Question 4 and 5**
>
> ***Q4.Could this framework be adapted to nonlinear merging or adapter-based composition (e.g., LoRA fusion)?***
>
> The key advantage of our framework comes from the low-dimensional parameterisation of the downstream learner, rather than from any specific linearity assumption on the merge itself. As long as the adaptation can be expressed through a relatively small set of tunable parameters and we can place a prior/posterior distribution over them, the same PAC-Bayes analysis applies.
>
> This includes nonlinear merging schemes and adapter-based compositions such as LoRA fusion. In such cases, the merge “parameters” are the coefficients that combine or gate the underlying adapters or submodules. If this parameter space remains low-dimensional, the KL term stays controlled and the resulting bound can remain non-vacuous even for large base models.
>
> To illustrate this, we applied our framework to LoRAHub (Huang et al., 2023), which is an adapter fusion method, with Flan-T5-Large on four Big-Bench Hard tasks: Logical Deduction with 3 objects, Logical Deduction with 5 objects, Logical Deduction with 7 objects, and Hyperbaton. We compare the vanilla LoRAHub optimisation, the bound-optimised version, and the data-dependent prior (DDP) variant:
> | Task | Vanilla (Train/Test/Bound) | Bound Opt (Train/Test/Bound) | DDP (Train/Test/Bound) |
> |------|-----------------------------|-------------------------------|-------------------------|
> | LD(5) | 0.5687 / 0.6298 / 0.9662 | 0.5333 / 0.6444 / 0.8912 | 0.5680 / 0.6293 / 0.7573 |
> | LD(3) | 0.4767 / 0.5116 / 0.9555 | 0.4427 / 0.5284 / 0.8499 | 0.4267 / 0.4622 / 0.6501 |
> | LD(7) | 0.5887 / 0.5929 / 0.9129 | 0.5767 / 0.6178 / 0.9095 | 0.5780 / 0.5920 / 0.7692 |
> | Hyperbaton | 0.2853 / 0.3524 / 0.8563 | 0.2327 / 0.3036 / 0.8133 | 0.2427 / 0.3284 / 0.4578 |
>
> These results show that adapter-based composition is fully compatible with our PAC-Bayesian certification scheme. In particular, the DDP variant substantially tightens the bound while keeping test performance comparable, confirming that the framework extends beyond linear model merging to more general low-dimensional fusion mechanisms such as LoRAHub.
>
> ***Q5. How would the results compare if the merging coefficients were learned jointly with fine-tuning of small subsets of parameters?***
>
> Fine-tuning even small subsets of parameters substantially increases the dimensionality of the learning problem, which directly affects the PAC-Bayesian KL term. In our setting, the smallest model is CLIP ViT-B/32, which already has around 88M parameters. Even if only a very small portion is tuned, the number of updated parameters typically exceeds tens of thousands. This scale is already too large to yield non-vacuous bounds in mid-shot or low-shot regimes, regardless of the optimisation strategy.
>
> This observation is consistent with prior work. As noted in L115–116 and L369–370, Lotfi et al. (2024) obtain non-vacuous PAC-Bayes bounds for SubLoRA, which is a compressed, low-rank variant of LoRA. However, their guarantees require tens of thousands of examples, and the compression step is essential precisely because full-rank or lightly-restricted LoRA still introduces too many learnable parameters. In our few-shot setting, even SubLoRA fails to produce non-vacuous bounds, indicating that further reducing the parameter dimension is necessary.
>
> Model merging naturally provides such an extreme reduction. The number of tunable parameters collapses to a handful of merge coefficients, and this low-dimensional structure is what allows the certification to remain non-vacuous with only tens or hundreds of examples. Jointly learning merge coefficients and fine-tuned parameters would move the method back into a high-dimensional regime where PAC-Bayesian bounds become vacuous.

---

> > ### Comment · Reviewer_VEYW · 2025-11-28
> >
> > I'd like to thank the authors for their really detailed response, which clarifies all my questions! Although the issue of "limited theoretical depth" remains, the adaptation of PAC-Bayes theory on these applications is certainly interesting. I'll re-read the paper along with the authors' comments here and the supplementary material, and update my score if warranted/possible (there are some Open-review issues at the moment..). Thanks again to all authors!

---

### Official Review · Reviewer_Dkxa · 2025-11-01

**Soundness:** 3
**Presentation:** 2
**Contribution:** 2
**Rating:** 4
**Confidence:** 2

**Summary:**

This paper reveals that model merging methods such as Task Arithmetic, Ties-Merging, and AdaMerging can naturally yield non-vacuous PAC-Bayesian generalization guarantees, even for large models like ViT-B and Mistral-7B trained on as few as 100 examples. By viewing the small set of fusion weights as the learnable parameters, the approach sharply reduces effective model capacity, tightening PAC-Bayes bounds. The authors further enhance this link through bound-optimized objectives and data-dependent priors, demonstrating empirically that merging-based few-shot adaptation can be both practical and theoretically certifiable-a first for large-scale networks in low-data regimes.

**Strengths:**

1. The author demonstrated that certifiable generalization on ViT-B and Mistral-7B under 100-shot learning is unprecedented, compared with previous attempts (e.g., Lotfi et al. 2024) which require thousands of samples or smaller models.
2. Comprehensive experiments span both vision and language tasks (EuroSAT, GTSRB, MNIST, DTD, BBH, TweetEval), showing consistent trends across settings.

**Weaknesses:**

1. The introduction jumps quickly into technical framing (generalisation bounds, PAC-Bayes, model fusion) without clearly articulating the conceptual motivation: Why should we care about certifying large models in the first place?
2. The work did not conduct sensitivity experiment on prior variance $\lambda$ or the confidence parameter $\delta$.

**Questions:**

1. The guarantees apply only to IID settings. I am curious how the model behaves under a realistic adversarial attacks like PGD attack[1].
2. Could PAC-Bayesian interpretation also apply to other parameter-efficient tuning methods like LoRA? It would be helpful to know if model merging is uniquely suitable for certification, or if low-dimensionality alone explains the effect.
3. The experiments focus on 100-shot learning. would the certification remain valid (or meaningful) under more extreme few-shot scenarios, such as 10-shot or 1-shot?

[1] Madry, Aleksander, et al. "Towards deep learning models resistant to adversarial attacks." arXiv preprint arXiv:1706.06083 (2017).

---

> ### Author Response · Authors · 2025-11-20
> **Answers to Weaknesses**
>
> We thank the reviewer for the time and effort spent carefully evaluating our work. The following responses address the concerns raised. Please let us know if any further clarification is needed.
>
> ***W1. The introduction jumps quickly into technical framing (generalisation bounds, PAC-Bayes, model fusion) without clearly articulating the conceptual motivation: Why should we care about certifying large models in the first place?***
>
> Certification is important whenever a model is used in high-stakes settings such as safety, medicine, security & defense, autonomous vehicles, etc. In such high-stakes settings, guarantees that models will generalise are required for stakeholders to trust AI for deployment. This applies to all models, not only large ones.
>
> The problem is that, in practice, large modern models have been considered almost impossible to certify, especially when combined with small training sets. Existing generalisation bounds scale poorly with the size of the model, so applying them to models with millions or billions of parameters trained on small datasets usually produces vacuous guarantees. As a result, certification has been out of reach for large models that are empirically the most effective, limiting our ability to deploy them in high-stakes scenarios.
>
> Our work points out that model merging offers a way around this. Model merging is widely used by practitioners, and it parametrises downstream learning using only a small number of merge coefficients rather than the full parameter set of the underlying models. Because the number of learnable parameters is small, the generalisation bounds become meaningful even with limited data. This makes certification feasible for large models in situations where conventional certificates are vacuous.
>
> We will revise the introduction so that this motivation is stated more clearly before introducing the technical details.
>
> ***W2. The work did not conduct sensitivity experiment on prior variance  or the confidence parameter.***
>
> The sensitivity analysis with respect to the prior/posterior variance is presented in Table 7 and Appendix C.3. As shown there, the certification results are stable across diverse variance settings.
>
> Regarding the role of the confidence level, $\delta$ is not a tunable hyperparameter for optimisation but rather a user-specified confidence requirement, as in standard PAC-Bayes formulations. Once $\delta$ is fixed, it contributes only through the term involving $\log\frac{n}{\delta}$, which shifts all bounds by the same constant amount and does not interact with the optimisation of the posterior. For this reason, changing $\delta$ does not meaningfully alter the conclusions about variance sensitivity, and it is not something our method attempts to optimise.
>
> Overall, Table 7 indicates that the proposed certification approach does not rely on delicate tuning of the prior/posterior variance, and the behaviour of the bound is consistent across the tested settings.

---

> ### Author Response · Authors · 2025-11-20
> **Answers to Questions**
>
> ***Q1. The guarantees apply only to IID settings. I am curious how the model behaves under a realistic adversarial attacks like PGD attack***
>
> This work focuses on certifying IID generalisation, so robustness to adversarial attacks such as PGD is outside the main scope of the paper. However, the PAC-Bayes framework we use is not restricted to the standard supervised loss. The bound applies to any loss taking values in $[0,1]$, so we can instantiate the same certificate for adversarial risk simply by redefining the loss.
>
> For an $\\epsilon$-bounded perturbation set, define the adversarial loss:
> $$
> \\ell\_{\\mathrm{adv}}(f\_\\theta, x, y)
> = \\max\_{\\|\\delta\\| \\le \\epsilon} \\, \\ell(f\_\\theta(x + \\delta), y).
> $$
>
> The corresponding empirical and population adversarial risks are:
> $$
> \\hat{L}\_{\\mathrm{adv}}(Q)
> = \\frac{1}{n} \\sum\_{i=1}^n
> \\mathbb{E}\_{\\theta \\sim Q}\\left[\\ell\_{\\mathrm{adv}}(f\_\\theta, x\_i, y\_i)\\right],
> $$
>
> $$
> L\_{\\mathrm{adv}}(Q)
> = \\mathbb{E}\_{(x,y) \\sim \\mathcal{D}}
> \\mathbb{E}\_{\\theta \\sim Q}\\left[\\ell\_{\\mathrm{adv}}(f\_\\theta, x, y)\\right].
> $$
>
> Since $\\ell\_{\\mathrm{adv}} \\in [0,1]$, the Langford--Seeger PAC-Bayes bound applies without modification:
> $$
> \\mathrm{kl}\\!\\left(\\hat{L}\_{\\mathrm{adv}}(Q) \\,\\|\\, L\_{\\mathrm{adv}}(Q)\\right)
> \\le
> \\frac{\\mathrm{KL}(Q\\|P) + \\log(n/\\delta)}{n - 1}.
> $$
>
> Thus, the same certificate holds for adversarial performance; the only practical change is that $\\hat{L}(Q)$ is computed using adversarial loss (e.g., via PGD).
>
>
>  We present certificates regarding the adversarial generalisation error with Taskwise Adamerge on MNIST below:
> | Method              | Metric      | PGD-MNIST |
> |---------------------|------------|-----------|
> | LW-Ada + DDP        | Test Error | 0.249     |
> |                     | Train Error| 0.176     |
> |                     | PB Bound   | 0.438     |
> |                     | Upper Bound| 0.454     |
>
> For adversarial attack, we utilise PGD attack with step number 7, step size 2/255, threshold 8/255.
>
>
> ***Q2. Could PAC-Bayesian interpretation also apply to other parameter-efficient tuning methods like LoRA? It would be helpful to know if model merging is uniquely suitable for certification, or if low-dimensionality alone explains the effect.***
>
> As noted in L115–116 and L369–370, there is a prior work showing that PAC-Bayesian interpretations can apply to certain LoRA-style parameter-efficient tuning methods. In particular, Lotfi et al. (2024) obtain non-vacuous bounds for large models using SubLoRA, which is a compressed subspace version of LoRA. However, their results rely on tens of thousands of training examples to obtain non-vacuous bounds, even after utilising SubLoRA compression.
>
> For the mid-shot and low-shot settings considered in our work, even SubLoRA updates tend to introduce too many learnable parameters to yield non-vacuous bounds. In these regimes, a more compact parameterisation is required. Model merging provides this naturally: it can be viewed as an extreme form of PEFT where the number of tunable parameters is reduced to just a few merge coefficients. This low-dimensional structure is what makes certification feasible with only a small amount of data.
>
> ***Q3. The experiments focus on 100-shot learning. would the certification remain valid (or meaningful) under more extreme few-shot scenarios, such as 10-shot or 1-shot?***
>
> Figure 3 presents results where the number of training examples is reduced down to 40. It shows that using our data-dependent prior approach, some non-vacuous certificates can still be obtained.
> We now also provide additional 10-shot results below.
>
> | Method                     | Metric       | eurosat | gtsrb | svhn | mnist | dtd  |
> |---------------------------|--------------|---------|-------|------|-------|------|
> | TW-Ada + Bound Opt.       | Test Error   | 0.438   | 0.651 | 0.370 | 0.167 | 0.564 |
> |                           | Train Error  | 0.560   | 0.280 | 0.240 | 0.260 | 0.020 |
> |                           | PB Bound     | 1.000   | 0.834 | 0.870 | 0.876 | 0.597 |
> |                           | Upper Bound  | 1.149   | 0.892 | 0.959 | 0.969 | 0.653 |
>
> In this extreme low-shot regime, we run into an inherent limitation in PAC-Bayes bounds (in fact, any generalisation theory), that is independent of model complexity. The generalisation term includes  $\log(n/\delta)/{n}$ which becomes large when n is very small. This behaviour occurs even if the effective parameter dimension is close to zero, so vacuous bounds are unavoidable in the extreme few-shot 1-10 shot regime.
>
> Nevertheless, it’s worth emphasising that we have already reduced state of the art in terms of certifiable training set size by about two orders of magnitude from 10,000s (Lotfi 2024) to around 100 (This paper). This is now very close to the best possible (around 10 shot), as explained above.

---

> > ### Comment · Reviewer_Dkxa · 2025-11-26
> >
> > Thanks the author for the rebuttal and clarifications. The observation that merging-based adaptation yields certifiable generalisation in low-shot settings for large models is interesting. The additional explanations helped me better understand the core contribution of the paper, especially why model merging provides a low-dimensional parameter space where PAC-Bayes certificates can become non-vacuous. I have increased my score from 4 to 6.
> >
> > I still have some uncertainty about how widely model merging is used in real-world pipelines and about the broader prevalence of PAC-Bayes certificates in practical deployment settings. Given this, my confidence score remains at 2, as I am not an expert in this specific line of work and would give the decision to other reviewers.

---

> > > ### Author Response · Authors · 2025-11-27
> > >
> > > We sincerely thank the reviewer for the careful reading of both the manuscript and the rebuttal. If any further questions arise, we would be glad to address them.
> > >
> > > We also provide a few additional clarifications that may help address the remaining concerns regarding the practical relevance of model merging and PAC-Bayesian certification.
> > >
> > > ***Model merging in practice***
> > >
> > > As summarised in [R1], model merging has rapidly emerged as a practical tool for large-scale LLM and MLLM training, with demonstrated use in generative modeling, continual learning, multi-task learning, domain generalisation, federated learning, and few-shot learning. Furthermore, [R2] states  that many top entries on the Open LLM Leaderboard are the result of merging-based pipelines. These results indicate that model merging is no longer limited to research-only settings; it is increasingly a standard technique for deploying and updating large models in real-world workflows.
> > >
> > > ***PAC-Bayes certificates in practical settings***
> > >
> > > PAC-Bayesian analysis has also seen wide adoption across multiple practical domains. Recent works apply PAC-Bayes to domain adaptation [R3], meta learning [R4], multi-task learning [R5] and generative modeling [R6, R7], highlighting its usefulness for understanding and certifying generalisation in diverse application areas. Moreover, several recent papers provide non-vacuous PAC-Bayesian bounds for large-scale or practical setups such as LLM training using large-scale datasets [R8, R9]. These developments show that PAC-Bayes is increasingly viewed as a practical tool for obtaining reliable guarantees in modern machine-learning pipelines. Our contribution is to help establish PAC-bayes certification in the low-shot regime with large-scale models that are used in modern days.
> > >
> > > ***Summary***
> > >
> > > Overall, both model merging and PAC-Bayesian certification have become significantly more prevalent in real-world ML practice than in earlier years. Our work leverages this convergence: showing that the compact parameterisation inherent in modern merging methods makes PAC-Bayesian certification feasible even for very large models under low-shot conditions.
> > >
> > > [R1] Yang, Enneng, et al. "Model merging in llms, mllms, and beyond: Methods, theories, applications and opportunities." arXiv preprint arXiv:2408.07666, 2024
> > >
> > > [R2] Akiba, Takuya, et al. "Evolutionary optimization of model merging recipes." Nature Machine Intelligence 7.2 (2025): 195-204.
> > >
> > > [R3] Sicilia, Anthony, et al. "Pac-bayesian domain adaptation bounds for multiclass learners." UAI, 2022.
> > >
> > > [R4] Zakerinia, Hossein, Amin Behjati, and Christoph H. Lampert. "More flexible PAC-Bayesian meta-learning by learning learning algorithms." arXiv preprint arXiv:2402.04054 (2024).
> > >
> > > [R5] Zakerinia, Hossein, Dorsa Ghobadi, and Christoph H. Lampert. "Deep Multi-Task Learning Has Low Amortized Intrinsic Dimensionality." arXiv e-prints (2025): arXiv-2501.
> > >
> > > [R6] Mbacke, Sokhna Diarra, Florence Clerc, and Pascal Germain. "PAC-Bayesian generalization bounds for adversarial generative models." ICML, 2023.
> > >
> > > [R7] Mbacke, Sokhna Diarra, Florence Clerc, and Pascal Germain. "Statistical guarantees for variational autoencoders using pac-bayesian theory." NeurIPS, 2023
> > >
> > > [R8] Lotfi, Sanae, et al. "Non-Vacuous Generalization Bounds for Large Language Models." ICML, 2024.
> > >
> > > [R9] Lotfi, Sanae, et al. "Unlocking tokens as data points for generalization bounds on larger language models." NeurIPS, 2024.

---

### Meta-Review · Area_Chair_dSXm · 2026-01-07

**Summary:**

This paper reveals that model merging can yield non-vacuous PAC-Bayesian generalization guarantees for large models with a small number of samples. This is achieved by viewing the small set of fusion weights as the learnable parameters and the generalization bounds seem to no longer depend on the scale of the underlying models. The authors also demonstrated such bounds on many existing off-the-shelf model merging algorithms, with relatively large models under few-shot settings. Reviewers commonly agree that this is an interesting finding.

However, the technical contribution is kind of small -- no new generalization bounds were derived, and most results were reinterpreting existing theory. The paper basically says that when existing theory is applied to a setting with a small number of parameters, non-vacuous generalization bounds can be obtained (which directly follows existing works), and model merging is such a case with only a few parameters (trivial). I also find that the paper is presented in a misleading way that overstates its contributions -- the authors claimed this work is “the first non-vacuous certification for large models in low-shot settings” and compared to prior works such as Lotfi et al. 2024 which required many more examples or smaller models; however, certification in Lotfi et al. 2024 is end-to-end (starting from pre-training the model by themselves), while this paper has ignored how models are trained before merging and any generalization gaps from the training of the underlying models, which makes the certification here technically trivial. Additionally, there is also a common concern that the motivation and methodological explanation are unclear, as mentioned by multiple reviewers. I agree that the writing is unclear and confusing, even after reading both the paper and rebuttal, and I think some confusion is partly from that the authors overstated their contributions on handling large models and did not state upfront that their generalization bounds are not end-to-end.

Overall, I think this paper is below the acceptance threshold.

**Reviewer Concerns:**

There are outstanding major concerns on the limited technical contributions, as well as unclear motivation and lack of justification for the theory (see above).

Some other concerns have been addressed in the rebuttal, such as sensitivity results, difference compared to formal verification, overhead, and confusion with PAC-learning.

**Reviewer Scores:**

Reviewer xkfc rated 6, and they acknowledged the rebuttal and maintained their original score.

Reviewer Dkxa initially rated 4 and explicitly commented that they had increased their score to 6.

Reviewer VEYW initially rated 4 and acknowledged that the rebuttal clarified all their questions. However, they did not explicitly say if they would increase the rating (“I'll re-read the paper along with the authors' comments here and the supplementary material, and update my score if warranted/possible”). Some concerns, such as lack of methodological explanation and technical depth, remain there. The reviewer may maintain their score of 4.

Reviewer LQRu rated 4. Some of their concerns seem to be misunderstanding, but I think the concern on the lack of theoretical justification or intuitive insight remains. The reviewer may maintain their score of 4.

---

### Decision · Program_Chairs · 2026-01-26

Reject